# Mechanics and dynamics of pinning points on the Shirase Coast, West Antarctica

Holly Still[1] and Christina Hulbe[1]

[1]National School of Surveying, University of Otago, Dunedin, New Zealand

**Correspondence:** Holly Still (holly.still@otago.ac.nz)

**Abstract.** Ice rises and rumples, sites of localised ice-shelf grounding, modify ice-shelf flow by generating lateral and basal shear stresses, upstream compression and downstream tension. Studies of pinning points typically quantify this role indirectly, through related metrics such as a buttressing number. Here, we quantify the dynamic effects of pinning points directly, by comparing model-simulated stress states in the Ross Ice Shelf (RIS) with and without a specific set of pinning points located downstream of the MacAyeal and Bindschadler Ice Streams (MacIS and BIS, respectively). Because ice properties are only known indirectly, the experiment is repeated with different realisations of the ice softness. While longitudinal stretching, and thus ice velocity, is smaller with the pinning points, flow resistance generated by other grounded features is also smaller. Conversely, flow resistance generated by other grounded features increases when the pinning points are absent, providing a non-local control on the net effect of the pinning points on ice-shelf flow. We find that an ice stream located directly upstream of the pinning points, MacIS, is less responsive to their removal than the obliquely oriented BIS. This response is due to zones of locally higher basal drag acting on MacIS, which may itself be a consequence of the coupled ice-shelf and ice-stream response to the pinning points. We also find that inversion of present-day flow and thickness for basal friction and ice softness, without feature-specific, *a posteriori* adjustment, leads to the incorrect representation of ice rumple morphology and an incorrect boundary condition at the ice base. Viewed from the perspective of change detection, we find that following pinning point removal, the ice shelf undergoes an adjustment to a new steady-state that involves an initial increase in ice speeds across the eastern ice shelf, followed by decaying flow speeds, as mass flux reduces thickness gradients in some areas and increases thickness gradients in others. Increases in ice-stream flow speeds persist with no further adjustment, even without sustained grounding-line retreat. Where pinning point effects are important, model tuning that respects their morphology is necessary to represent the system as a whole and inform interpretations of observed change.

## 1 Introduction

Ice shelves regulate the Antarctic contribution to sea level rise via their influence on grounding-line position and tributary glacier dynamics. Individual ice shelves are regulated by environmental conditions including their geographic setting. An ice shelf laterally confined within an embayment experiences reduced longitudinal tensile stress (and stretching) relative to an unconfined ice shelf due to lateral shearing where ice flows past coastal features and islands (Sanderson, 1979; Haseloff and Sergienko, 2018; Pegler, 2018). Where floating ice runs aground, a pinning point forms and resulting compression and

shearing further reduce longitudinal stresses (Favier et al., 2012; Borstad et al., 2013; Favier and Pattyn, 2015; Berger et al., 2016). Altogether, the rate of mass flux is moderated in an effect commonly referred to as 'buttressing' exerted on upstream grounded ice by an ice shelf (Dupont and Alley, 2005, 2006; Gudmundsson, 2013; Fürst et al., 2016).

By generating resistive stresses, pinning points modify the velocity pattern and, via advection, the thickness pattern of an ice shelf. The ice-shelf momentum balance is "non-local" due to very low basal traction (Thomas, 1979), and changes to stresses in an ice shelf may propagate across the grounding line to low basal-traction, tributary ice streams (Reese et al., 2018). The momentum and mass perturbations together must be balanced by changes in thickness and resistive stresses elsewhere in the ice-shelf and ice-sheet system. Enhanced deformation around a pinning point also affects ice properties such as ice crystal fabric and temperature, modifying softness and thus ice flow (e.g., Borstad et al., 2013). Altogether, the ice shelf is a coupled system in which a change in any specific location may, though the momentum and mass balances, drive change in resistive stresses and ice thickness elsewhere. The aim of the present work is to quantify the complete system of mass and momentum adjustments caused by a specific set of pinning points in the Ross Ice Shelf (RIS), West Antarctica.

The importance of pinning points to ice-shelf stability and grounding-line position has been widely examined observationally (e.g., Matsuoka et al., 2015). For example, the speed-up and grounding-line retreat of Pine Island Glacier following the loss of a pinning point has been documented by Bindschadler (2002), Rignot (2002), Jenkins et al. (2010) and Arndt et al. (2018). The individual force balance contributions of various Antarctic pinning points have been computed from observational data by Thomas (1973), Thomas (1979), Thomas and MacAyeal (1982), MacAyeal et al. (1987) and Still et al. (2019), but such calculations cannot address non-local effects. Flow buttressing numbers (Borstad et al., 2013; Fürst et al., 2016) provide a summary view of the non-local effects but do not quantify the pinning point contribution to individual resistive stresses.

Observational-data driven analysis provides snapshot-like, summary views of recent conditions but cannot address how the coupled system would adjust to changes in individual features. More theoretical approaches examine coupled mass and momentum effects of pinning points across the interconnected ice-shelf and ice-sheet system. For example, Goldberg et al. (2009) conducted idealised simulations of grounding-line position and mechanics with (and without) an ice rise to demonstrate how an ice rise can modify vulnerability to the marine ice-sheet instability. Favier et al. (2012) demonstrated that local changes to ice thickness due to the emergence of a pinning point generate feedbacks in the stress balance that can maintain the local thickness perturbation and thus the grounded feature. Fried et al. (2014) examined the emergence of ice rises in the RIS as a source of thickness transients that drove past grounding-line transgression. Nias et al. (2016) simulated the Thwaites Glacier response to changing contact with a pinning point beneath its floating ice tongue and concluded that basal traction on the grounded ice was more important to the glacier response than the direct mechanical effects of the pinning point itself.

We take a different perspective to examine the dynamical role of pinning points that involves a detailed analysis of the stress patterns across the RIS. Our aim is to quantify the complete pinning point contribution to ice dynamics, and we do this by performing numerical model simulations of ice-shelf and ice-stream flow with and without a collection of lightly grounded, low relief ice rumples (hereafter called the Shirase Coast Ice Rumples, SCIR). Differences between two steady states – with and without the SCIR – show how the coupled system responds to their presence, including a repartitioning of resistive stresses

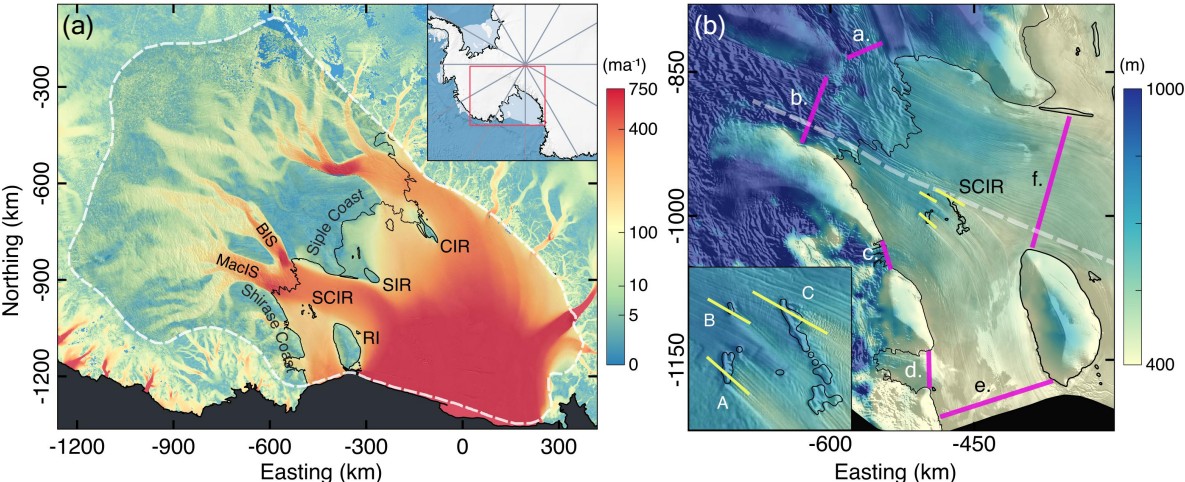

**Figure 1.** Pinning points in the RIS. In panel (a), large pinning points are labelled: SCIR = the Shirase Coast Ice Rumples, RI = Roosevelt Island, SIR = Steershead Ice Rise and CIR = Crary Ice Rise. The colour map of surface ice velocity magnitude is from the MEaSURES velocity dataset (Rignot et al., 2011a). The black line indicates the grounding lines (Bindschadler et al., 2011) and the white dashed line is the limit of the finite element model domain. Panel (b) shows the along-flow cross-sections intersecting the SCIR in Figs. 3 and 10 (yellow and white dashed lines), and the gates used for mass flux calculations in Table 3. The colour map of ice thickness is from the Bedmap2 compilation (Fretwell et al., 2013). In each figure from hereon, datasets are mapped with a Polar Stereographic Projection with a central meridian of $0°$ and a standard latitude of $71°$S, and in most cases, overlayed onto the MODIS MOA (Haran et al., 2014).

and a redistribution of ice mass. Properties of the ice and subglacial bed, which must be inferred during model initialisation, are also examined using this approach.

The flow of the RIS is regulated by a diverse collection of ice rises and rumples, the surface morphological expressions of ice shelf flow over or around pinning points (Fig. 1). Ice rises have a distinct dome-shaped morphology with ice shelf flow diverging around the rise, while undulating ice rumples form where the ice continues to flow directly over the area of

localised grounding (Martin and Sanderson, 1980; Matsuoka et al., 2015). Large ice rises in the RIS include Crary Ice Rise, Roosevelt Island and Steershead Ice Rise. The SCIR are a group of smaller, unnamed ice rumples located along the Siple and Shirase Coasts. While individual rumples in the SCIR are relatively small, the lightly–grounded and low basal traction features collectively generate flow resistance comparable to larger ice rises in the RIS (Crary Ice Rise, Steershead Ice Rise and Roosevelt Island) (Still et al., 2019). The SCIR are located 60 km downstream from the grounding lines of MacAyeal

and Bindschadler Ice Streams (MacIS and BIS), two large outlet streams of the West Antarctic Ice Sheet, and within a cove upstream of Roosevelt Island. This setting stands out as a region of significant buttressing in the RIS (Reese et al., 2018) and the arrangement allows the mass and momentum balances to be examined in a non-simple geometry.

## 2  Method

### 2.1  Ice sheet model

ISSM is an open-source, finite-element ice flow model that solves the conservation equations for mass and momentum in combination with appropriate boundary conditions and the constitutive relationship for ice (Larour et al., 2012). The Shallow Shelf (or shelfy-stream) Approximation (SSA) (Morland, 1987; MacAyeal, 1989) of the full-Stokes equations, appropriate for sliding over a very weak substrate such as water or water-saturated subglacial till, is used to simulate ice-shelf and ice-sheet flow. The fast flowing RIS ice streams draining the West Antarctic Ice Sheet are characterised by thawed bases and significant

sliding over soft subglacial till (MacAyeal et al., 1995; Joughin et al., 2004), justifying the SSA for the present study.

The model domain includes the RIS and its West Antarctic catchment (Fig. 1a). To the west of the RIS, the domain is bounded by the coastline and glacier outlets of the Transantarctic Mountains. The initial grounding-line position is defined by the Bedmap2 grounded ice mask (Fretwell et al., 2013) and the ice-shelf front is fixed at the ice front position in the 2008-2009 MODIS Mosaic of Antarctica (Haran et al., 2014). The SSA equations are solved on an unstructured triangular mesh with

anisotropic mesh refinement (Larour et al., 2012) applied to achieve a fine spatial resolution over features of interest and across transitions in stress boundary conditions. Mesh element size ranges from approximately 1 km edge lengths over pinning points, ice-stream shear margins and the grounding line, to approximately 10 km edge lengths over the inland ice sheet and central ice shelf. All together, there are 270 000 elements (134 000 vertices) in the model mesh.

The momentum balance requires a constitutive law describing the nonlinear relationship between stress and strain rate (Glen,

90  1955)

$$\tau_{ij} = 2\eta\dot{\epsilon}_{ij} \tag{1}$$

where $\tau_{ij}$ is the deviatoric stress tensor and $\dot{\epsilon}_{ij}$ is the strain rate tensor. In the SSA, shearing between horizontal planes is zero and therefore a depth-averaged effective viscosity

$$\bar{\eta} = \frac{\bar{B}}{2\dot{\epsilon}_e^{1-1/n}} \tag{2}$$

where $\bar{B}$ is a depth-averaged rate factor (ice stiffness parameter) and $n = 3$ is used. The effective strain rate $\dot{\epsilon}_e$ is the second invariant of the strain rate tensor (Cuffey and Paterson, 2010, pg. 59). Spatial variations in $\bar{B}$ are a function of ice temperature and other material properties, none of which are represented explicitly in the present model. Instead, $\bar{B}$ is inferred from observational data.

Resistance to ice motion along the basal boundary is described by a linear friction law that relates basal shear stress $\tau_b$ to the

basal ice velocity

$$\tau_b = -\alpha^2 N \mathbf{u}_b \tag{3}$$

where $\alpha$ is the friction coefficient and $\mathbf{u}_b$ is the basal velocity. Without shearing between horizontal planes, the horizontal components of velocity do not vary with depth. The negative sign represents a traction opposing the direction of motion. The

coefficient $\alpha$ represents the mechanical and thermal properties of the ice/bed interface and the underlying subglacial material. The value of $\alpha$ is equal to zero where ice is afloat and is greater than zero at mesh elements where ice is grounded. $N$ is the basal effective water pressure, defined as the overburden pressure minus the water pressure at the ice-sheet base

$$N = g(\rho_i H + \rho_w z_b) \tag{4}$$

where $g$ is acceleration due to gravity, $\rho_i$ is the ice density, $\rho_w$ is the water density, $H$ is the ice thickness and $z_b$ is the bedrock elevation with respect to sea level (Budd et al., 1979; Bindschadler, 1983). The basal water system is assumed to be perfectly connected and $N$ approaches zero as ice goes afloat. The primary interest here is a comparison of two steady-state cases and neither details of the basal water system nor alternative representations of sliding are considered. Spatial variations in $\alpha$ are inferred from observational data.

A mass transport equation introduces time into the model. Conservation of mass is

$$\frac{\partial H}{\partial t} = -\nabla \cdot (\bar{\mathbf{u}}H) + \dot{a} - \dot{b} \tag{5}$$

where $\bar{\mathbf{u}}$ is the horizontal velocity, $\dot{a}$ is the surface accumulation rate and $\dot{b}$ is the basal accumulation rate. The basal accumulation rate is parameterised following Martin et al. (2011) and Beckmann and Goosse (2003)

$$\dot{b} = \rho_{sw} c_p \gamma_T F_{\text{melt}} (T_o - T_f)/(L_i \rho_i) \tag{6}$$

where $\rho_{sw}$ is the density of seawater, $c_p$ is the specific heat capacity of the ocean layer on which the ice floats, $\gamma_T$ is the thermal exchange velocity, $F_{\text{melt}}$ is a tunable constant, $T_o$ is the temperature of seawater beneath the ice shelf, $T_f$ is the freezing temperature of seawater at the depth of the ice shelf base and $L_i$ is the latent heat capacity of ice. Both grounding-line migration and the representation of basal friction for partially floating elements (as the grounding line migrates) are treated using the sub-element parameterisation scheme ('SEP2') of Seroussi et al. (2014). Physical constants used in the model are listed in the Supplement, Table S1.

## 2.2 Model initialisation

The experiment design requires steady-state reference and perturbed model configurations to quantify the net effect of the SCIR on the momentum and mass balances. Initial ice velocity and thickness are prescribed using recent satellite-derived observations. Surface velocities are from the 750 m grid-spacing Landsat 8 dataset (Fahnestock et al., 2016) and the 900 m grid-spacing MEaSUREs dataset (Rignot et al., 2011a) representing time periods from 2013 to 2016, and 2007 to 2009, respectively. The two velocity datasets are merged, with the MEaSUREs dataset used to fill the region beyond the Landsat 8 latitudinal limit. Ice thickness and the elevation of the subglacial topography and seafloor are from the 1 km grid-spacing Bedmap2 compilation (Fretwell et al., 2013). Surface mass balance (ice-equivalent accumulation rate) is prescribed according to Vaughan et al. (1999) and the basal melt rate parameter $F_{melt}$ (Martin et al., 2011) is tuned so that the grounding line remains within 50 km of its present-day position during reference model relaxation. Dirichlet conditions are imposed on the upstream boundaries of the

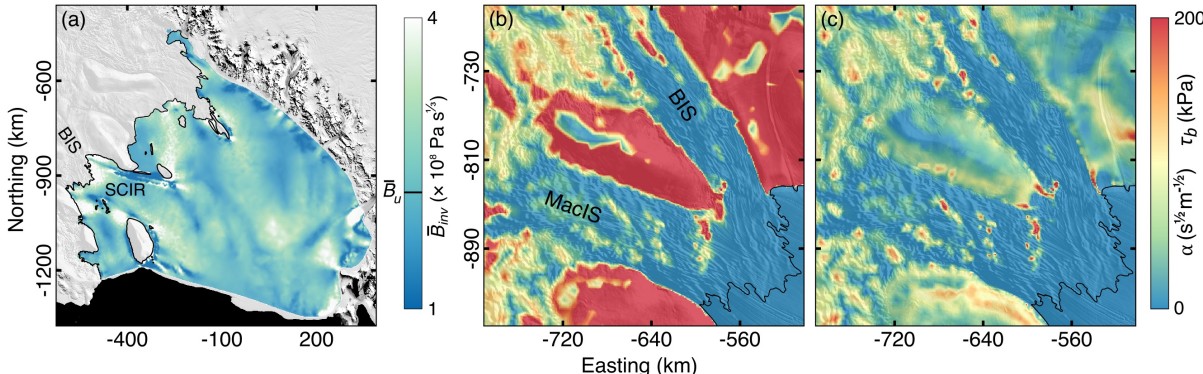

**Figure 2.** Panel (a) shows the inverse rate factor $\bar{B}_{inv}$ in the floating part of the model domain. The value of uniform $\bar{B}_u$ is shown in the colourbar of panel (a). Panels (b) and (c) focus on MacIS and BIS, showing the friction coefficient $\alpha$ and corresponding basal drag $\tau_b = -\alpha^2 N\mathbf{u}_b$. The grounding line is the observed position from Bindschadler et al. (2011).

model domain using observed velocity and ice thickness, and zero-slope Neumann conditions are specified on the downstream
(ice-shelf front) boundary.

The Bedmap2 bathymetry must be adjusted in order to ensure model fidelity to individual pinning points in the SCIR complex. To achieve this, pinning point perimeters are delineated using the MODIS Mosaic of Antarctica (MOA) (Haran et al., 2014) and Landsat 8 imagery (courtesy of the U.S. Geological Survey), and the surrounding bathymetry is excavated by 500 m to prohibit excess grounding during the model initialisation process (Fig. S1). Bathymetry within 50 km of the present-day
grounding line is left unmodified. Adjustments to the Bedmap2 bathymetry are described in the Supplement, Section 1.2.

An inverse method is used to infer the depth-averaged rate factor $\bar{B}$ and basal friction coefficient $\alpha$ from observed ice velocity and geometry (MacAyeal, 1993; Morlighem et al., 2010, 2013) (Figs. 2 and S3). The inverse method seeks to minimise the value of a cost function, a measure of the misfit between observed and modelled ice velocities, integrated over the whole model domain. Terms in the cost function include absolute and logarithmic misfits, and a third regularisation term is included to
prevent physically unrealistic variations in $\bar{B}$ and $\alpha$ over short spatial scales. The procedure is as follows:

1. **First inversion for $\bar{B}$ for floating and grounded ice**. The initial estimate is a uniform $\bar{B}$ of $1.6 \times 10^8$ Pa s$^{1/3}$, corresponding to an ice temperature of -16.7°C (Cuffey and Paterson, 2010, pg. 75). Uniform values of $\alpha = 200$ s$^{1/2}$ m$^{-1/2}$ for grounded ice and $\alpha = 0$ s$^{1/2}$ m$^{-1/2}$ for floating ice are specified. An $\alpha$ value of 200 s$^{1/2}$ m$^{-1/2}$ corresponds to $\tau_b$ between ~50 and 150 kPa, which is appropriate for relatively slow moving or stagnant ice but not for the fast-flowing
RIS ice streams.

2. **First inversion for $\alpha$**. The new, spatially variable, $\bar{B}$ from step 1 is used in an inversion to estimate spatially variable $\alpha$ for grounded ice.

3. **Second inversion to improve $\bar{B}$.** The parameter is reset to the original uniform value and inferred again, now using the spatially variable $\alpha$ from step 2. This improves the quality of the inversion near the grounding line (i.e., a smaller misfit between modelled and observed velocity).

4. **Second inversion to improve $\alpha$.** The parameter is reset to the original uniform values and the spatially variable $\bar{B}_{inv}$ from step 3 is used in a new inversion for $\alpha$.

The inferred friction parameter $\alpha$ (step 4) assigned to SCIR mesh elements is further manipulated to improve simulated ice rumple morphology (discussed in detail in Section 3.1). Later, a second model is initialised with a spatially uniform $\bar{B}_u$ of $2.2 \times 10^8$ Pa s$^{1/3}$ to assess how the inferred $\bar{B}_{inv}$ pattern affects model outcomes. The parameter fields are held constant after initialisation.

The aim of comparing two steady-state model configurations is to evaluate the net effect of the SCIR on the momentum and mass balances. The reference model is relaxed by iterating with fixed boundary conditions for 500 steps with a timestep of 2 years to remove non-physical spikes associated with inconsistencies between observed datasets. During relaxation, the MacIS and BIS grounding lines, upstream of the SCIR, remain in their approximate present-day positions while the grounding line between Steershead Ice Rise and Crary Ice Rise advances by 150 km over a shallow seafloor. After $\sim$100 years, the rate of change in ice-shelf volume is 0.01% per year and after $\sim$450 years, the rate of change in ice-shelf volume is 0.001% per year. While the steady-state reference model does not replicate the present-day situation exactly, velocity and ice thickness patterns across the ice shelf are preserved (see Supplement, Sections 1.3, 1.4 and Figs. S4 and S5). The relaxed steady-state can thus be considered representative of the present-day behaviour of the RIS, and appropriate for model experiments intended to resolve the flow-regulating effects of pinning points.

## 2.3 Experiment design

The system-wide mechanical and dynamical effects of the SCIR are quantified by comparing simulations of RIS and tributary ice stream flow with and without the SCIR included in the model domain. The steady-state reference model is perturbed by excavating the bathymetry beneath the SCIR to prevent mechanical contact between the ice and seafloor, and stepped forward for 150 years with a timestep of one year. By 150 years, the rate of change in ice shelf volume is <0.001 %, indicating that the model has reached a new steady-state. Two model states are then compared: (1) the 'reference model' of RIS flow with the SCIR complex in its present-day configuration; and (2) the 'perturbed model', 150 years after removal of the SCIR from the model domain. Differences in the stress regime between the two model states are considered in a flow-following, $(l, t)$, coordinate system, as changes in longitudinal tension/compression $\bar{R}_{ll}$, lateral shearing $\bar{R}_{lt}$, and transverse tension/compression $\bar{R}_{tt}$ (e.g., van der Veen and Whillans, 1989; Price et al., 2002; van der Veen et al., 2014; van der Veen, 2016).

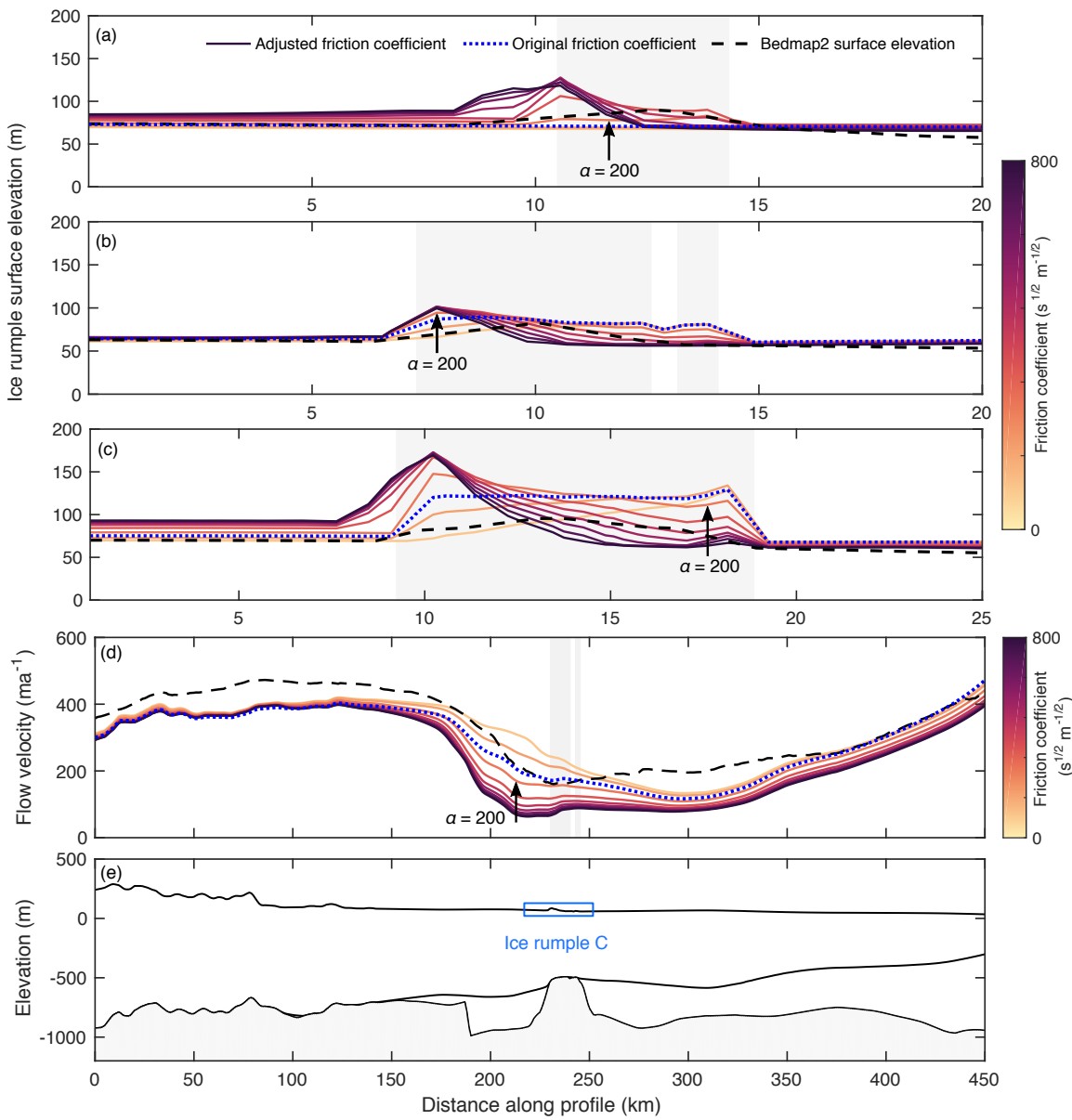

**Figure 3.** Surface morphology and ice velocity for different basal friction coefficient $\alpha$ values assigned to SCIR model nodes. Along-flow surface elevation profiles in panels (a), (b) and (c) demonstrate how selection of the friction coefficient before model relaxation affects ice thickness and surface elevation for three ice rumples in the SCIR complex (ice rumples A, B and C, respectively, see 1b for their location). Grey shaded boxes indicate model nodes where the ice shelf is grounded. Panel (d) demonstrates how selection of the friction coefficient affects the velocity magnitude. The profile in (d) represents a single pathway that begins 150 km upstream of the MacIS grounding line, intersects the SCIR rumple C, and ends at the shelf front. Panel (e) shows ice thickness and the underlying seafloor along this pathway in the reference model. The locations of the profiles in (a) to (e) are mapped in Fig. 1b.

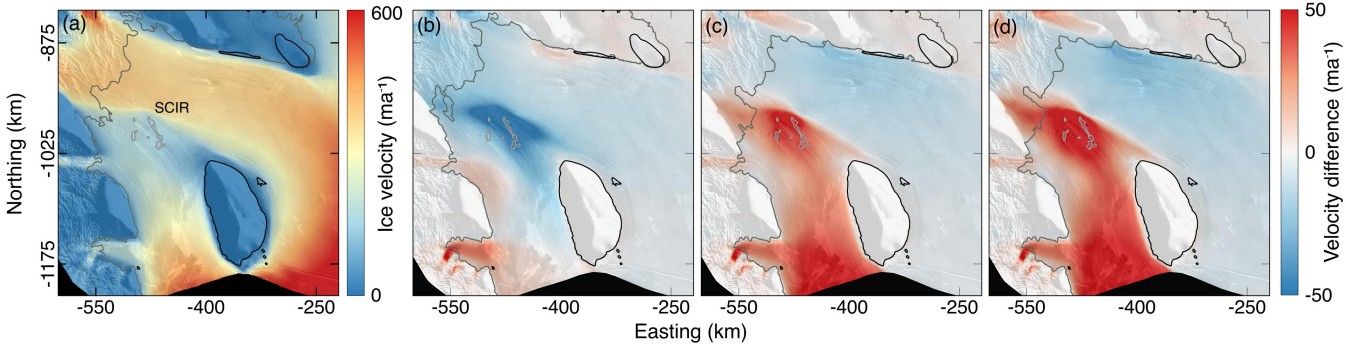

**Figure 4.** Ice velocity response to different friction coefficient values assigned to the SCIR model nodes. (a) Modelled ice velocity when $\alpha = 200 \text{ s}^{1/2} \text{ m}^{-1/2}$. (b-d) The difference in ice velocity between the reference model ($\alpha = 200 \text{ s}^{1/2} \text{ m}^{-1/2}$) and alternative relaxed models with varying $\alpha$ values assigned to the SCIR nodes before model relaxation. In (b), (c), and (d), $\alpha = 0$, 400, and 600 $\text{s}^{1/2} \text{ m}^{-1/2}$, respectively. The range $\alpha = 0$, 200, 400, 600 $\text{s}^{1/2} \text{ m}^{-1/2}$ corresponds to $\tau_b = 0$, 51, 67, 86 kPa (i.e., no basal traction or soft subglacial sediment, through to hard bedrock). Grounding-line positions are from the relaxed models with varying $\alpha$ values.

## 3 Results

### 3.1 The representation of ice rumples in the model

Friction coefficient values inferred for the SCIR using the typical model initialisation process, described above, do not yield
a realistic ice rumple geometry after the relaxation simulation. Excessively large $\alpha$ values (corresponding to $\tau_b > 400$ kPa) are inferred for mesh elements on the upstream side of the SCIR complex and zero values are inferred for downstream ice rumple elements (Fig. S2). While this arrangement broadly reproduces observed ice flow, it is inconsistent with the surface morphological expression and grounded area observed in satellite imagery. In particular, the smaller downstream pinning points in the complex have distinct surface relief that indicates larger than zero traction (Still et al., 2019). To address this problem,
the friction coefficient assigned to ice rumple nodes is manually adjusted before model relaxation to reproduce both observed ice flow and rumple morphology (Fig. 3).

Ice rumple geometry and morphology are an expression of the relationship between momentum and mass balances (Fig. 3). In general, when the friction coefficient (and thus basal drag) is larger, flow is slower over the upstream-most reaches of the SCIR, and ice is thicker upstream, thinner downstream and locally steeper than when the coefficient is smaller. While local
surface steepening helps to maintain ice flux past the obstacle, downstream thinning may reduce the extent of the model ice rumples. Conversely, a smaller friction coefficient generates more spatially extensive, lower amplitude features. The competing effects of basal drag and mass flux are most clearly observed for the relatively large SCIR rumple C (in Fig. 3), where larger $\alpha$ generates thickening and steepening along the upstream-most boundaries and thinning elsewhere. It is worth noting that there is an upper limit to these effects as velocity slows toward zero.

**Table 1.** Force budget components generated by the SCIR in the reference and perturbed models. Form drag $\mathbf{F}_f$ is the glaciostatic contribution to the net flow resistance, dynamic drag $\mathbf{F}_d$ is the viscous resistance associated with ice deformation around an obstacle, and the effective resistance $\mathbf{F}_e$ is the total reaction force arising from contact between the pinning point and the ice shelf base. The apparent basal shear stress $\tau_b$ is the quotient of $\mathbf{F}_e$ and the total pinning point grounded area (240.4 km$^2$). The equations to compute the force budget components are presented in the Supplement, Section 4.

| Model | | $\mathbf{F}_f$ | $\mathbf{F}_d$ | $\mathbf{F}_d/\mathbf{F}_f$ | $\mathbf{F}_e$ | $\tau_b$ |
|---|---|---|---|---|---|---|
| | | ($\times 10^{12}$ N) | ($\times 10^{12}$ N) | - | ($\times 10^{12}$ N) | (kPa) |
| $\bar{B}_{inv}$ | With SCIR | 44.8 | 1.6 | 0.04 | 7.2 | 30.0 |
| | No SCIR | 15.9 | 1.5 | 0.09 | 0.4 | 1.9 |
| $\bar{B}_u$ | With SCIR | 44.9 | 5.0 | 0.11 | 10.4 | 43.5 |
| | No SCIR | 11.0 | 1.2 | 0.11 | 0.2 | 0.8 |

Modelled and observed ice flow in the vicinity of the SCIR are comparable when the friction coefficient $\alpha = 200$ s$^{1/2}$ m$^{-1/2}$ is assigned to all grounded SCIR elements (Figs. 3 and 4). This $\alpha$ value corresponds to a mean $\tau_b$ of 50.3 kPa (Eqs. 3 and 4), which is similar to the value $\tau_b = 51.6$ kPa inferred via force budget analysis (Still et al., 2019). Varying the friction coefficient assigned to the SCIR mesh elements causes velocity to change over approximately 30% of the RIS (Fig. 4).

### 3.2   The rate factor $\bar{B}$

The magnitudes of resistive stresses depend in part on the rate factor $\bar{B}$ (Eq. 7), which in turn depends on ice properties. For example, fabric developed as ice deforms past an obstacle will modify how readily the ice deforms and thus resistive stress magnitudes near the obstacle. Model initialisation by inversion of present-day fields captures and represents the effects of ice properties as spatial variation in $\bar{B}_{inv}$ (Figs. 2a and S3). Without a parameterisation to update ice material properties, the spatial pattern is fixed to the model grid and its effects on ice deformation persist even after the SCIR are removed from the model

domain. A fixed $\bar{B}_{inv}$ pattern would be reasonable in a forward experiment, in which the immediate system response to loss of a pinning point is investigated, but may not be appropriate for other aims. With these issues in mind, the experiment is repeated using a spatially uniform $\bar{B}_u$ that best reproduces ice velocity across the grounding line upstream of the SCIR.

Resistive stress magnitudes generated by grounded features differ between the spatially variable $\bar{B}_{inv}$ and uniform $\bar{B}_u$ models (Table 1). As a consequence, the velocity fields required to transfer an equivalent amount of mass through the steady-

state ice-shelf system also differ. In general, $\bar{B}_u$ yields smaller mass flux upstream and larger mass flux downstream of the SCIR, relative to the $\bar{B}_{inv}$ model. In other words, the pinning points yield different effective resistance to ice flow in the $\bar{B}_{inv}$ and $\bar{B}_u$ cases (Table 1). These effects can be summarised using a force budget (Still et al., 2019) (Figs. S8 and S9). Relatively softer ice along the upstream margin of the SCIR complex in the $\bar{B}_{inv}$ case reduces dynamic drag around the ice rumples, relative to the $\bar{B}_u$ case (Fig. S8a and b), while the effect on the form drag, which reflects disturbance to the thickness field, is

negligible (Table 1). When the SCIR are removed from the model domain, the situation reverses and spatially variable $\bar{B}_{inv}$ leaves an imprint that is expressed as a difference in form drag (Table 1).

**Table 2.** Force budget components generated by Roosevelt Island in the reference and perturbed models. $\tau_b$ is not computed here because horizontal velocities on Roosevelt Island are nearly zero.

| Model | | $\mathbf{F}_f$ ($\times 10^{12}$ N) | $\mathbf{F}_d$ ($\times 10^{12}$ N) | $\mathbf{F}_d/\mathbf{F}_f$ - | $\mathbf{F}_e$ ($\times 10^{12}$ N) |
|---|---|---|---|---|---|
| $\bar{B}_{inv}$ | With SCIR | 136.4 | 19.9 | 0.15 | 37.8 |
| | No SCIR | 143.5 | 21.0 | 0.15 | 40.0 |
| $\bar{B}_u$ | With SCIR | 174.2 | 13.0 | 0.07 | 36.4 |
| | No SCIR | 195.0 | 13.3 | 0.07 | 39.2 |

The selection of $\bar{B}$ also affects flow resistance provided by Roosevelt Island. The difference in dynamic drag between the reference and perturbed models is larger in the $\bar{B}_{inv}$ case than in the $\bar{B}_u$ case. This is because the relatively more deformable ice along the margin of the island with $\bar{B}_{inv}$ participates in the regional re-partitioning of resistive stresses (Table 2). Differences in dynamic drag are observed all around the island, but are relatively large along the upstream edge and eastern side (Fig. S8). Along with this, the system experiences a smaller change in the (already lower) form drag around the island when ice stiffness is spatially variable rather than uniform (Table 2).

### 3.3  Stresses

The non-local nature of the ice shelf momentum balance is examined in detail by computing the pattern of resistive stresses acting on RIS flow. The gravitational driving stress $\tau_d$ must be balanced by resistive stresses including the longitudinal stress $R_{ll}$, transverse stress $R_{lt}$, and the lateral shear stress $R_{tt}$. Resistive stresses are computed using flow-following longitudinal $\dot{\epsilon}_{ll}$, transverse $\dot{\epsilon}_{tt}$ and shear $\dot{\epsilon}_{lt}$ strain rates from the model via Glen's flow law

$$\bar{R}_{ll} = \bar{B}\dot{\epsilon}_e^{\frac{1}{n}-1}(2\dot{\epsilon}_{ll} + \dot{\epsilon}_{tt})$$
$$\bar{R}_{tt} = \bar{B}\dot{\epsilon}_e^{\frac{1}{n}-1}(2\dot{\epsilon}_{tt} + \dot{\epsilon}_{ll}) \tag{7}$$
$$\bar{R}_{lt} = \bar{B}\dot{\epsilon}_e^{\frac{1}{n}-1}(\dot{\epsilon}_{lt}).$$

The strain rate components are computed as velocity gradients across elements in the model domain, in a flow-following coordinate system.

#### 3.3.1  The driving stress $\tau_d$

Pinning points generate locally large thickness gradients that in turn support the relatively high $\tau_d$ near the largest ice rumple (Fig. 5). Upstream of the SCIR, relatively thicker ice with a lower surface slope leads to lower driving stresses in comparison to the configuration without the SCIR. In the perturbed model, thinning upstream of the former SCIR results in a larger thickness gradient and locally larger driving stresses immediately downstream of the grounding lines of MacIS and BIS, and in some locations, the locally larger $\tau_d$ (on the order of 10 kPa) and mass flux are in line with grounding line retreat (Section 3.4).

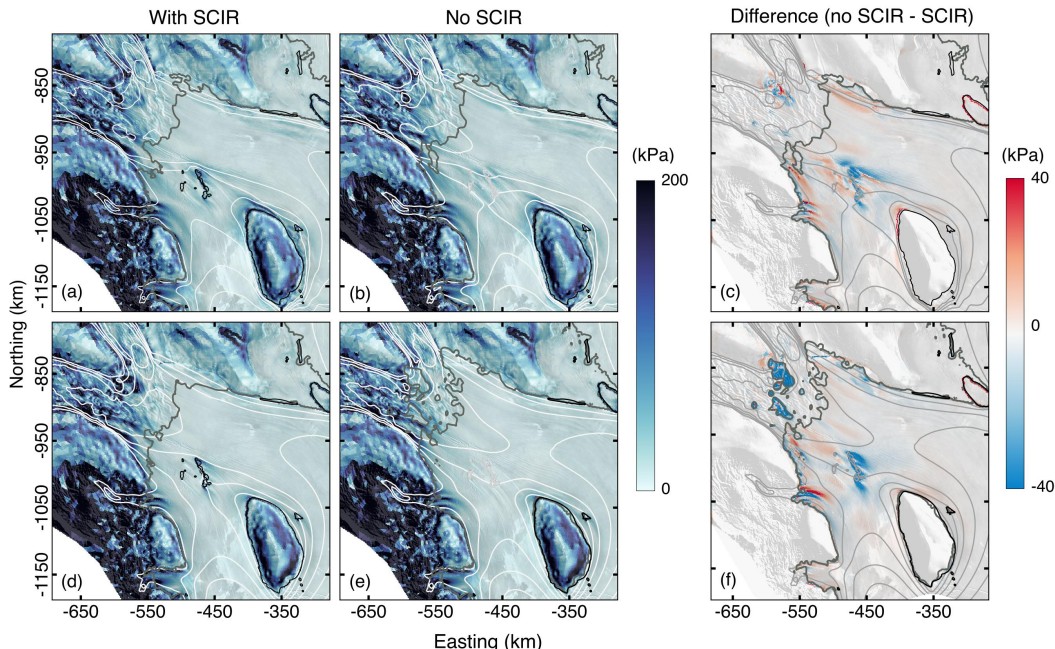

**Figure 5.** The gravitational driving stress $\tau_d$ acting on the RIS and tributary ice streams with and without the SCIR. In (a-c), the simulation is initialised with $\bar{B}_{inv}$. In (d-f), the simulation is initialised with $\bar{B}_u$. In (c) and (f), a positive (negative) change indicates an increase (decrease) in $\tau_d$ after removal of the SCIR. The velocity contour lines have an interval of $100\,\mathrm{ma}^{-1}$. Grounding-line positions in (a) and (d) are from the $\bar{B}_{inv}$ and $\bar{B}_u$ reference models (with SCIR). Grounding-line positions in (b) and (e) are from the $\bar{B}_{inv}$ and $\bar{B}_u$ perturbed models (without SCIR).

The SCIR have a more pronounced effect on driving stress immediately downstream of the grounding line in the $\bar{B}_u$ case, and
would have a similarly larger effect on flow buttressing in comparison to the $\bar{B}_{inv}$ case.

The SCIR may also affect $\tau_d$ upstream of the grounding line. Patches of relatively large decreases in driving stress (blue patches in Fig. 5f) coincide with local lows in the bed elevation, where thinner ice is predicted to go afloat in the perturbed model. Elsewhere, the differences between the two simulations are small, $<5$ kPa. The model does not simulate basal hydrology and $\alpha$ is held fixed, both of which may be variable and contribute to dynamic change in the driving stress.

### 3.3.2  Basal drag $\tau_b$

Differences in the basal friction parameter inferred during model initialisation between MacIS and BIS generate different patterns of basal drag $\tau_b$. Both ice streams are characterised by broad regions of very low flow resistance interrupted by 'sticky spots' with very high basal shear stress ($> 200$ kPa) (Fig. 2c). Sticky spots are localised areas of higher resistance to basal sliding associated with undulations in the subglacial topography, or variations in till properties and basal water pressure (Alley,
1993; Stokes et al., 2007). Model initialisation results in a greater density of sticky spots on MacIS than on BIS, a result that is

consistent with other inversions of observed velocities (Joughin et al., 2004; Sergienko et al., 2008), observations of ice stream surface morphology and textures (Stephenson and Bindschadler, 1990; Bindschadler and Scambos, 1991), and seismic surveys (Anandakrishnan and Alley, 1994; Luthra et al., 2016).

Sticky spot distribution conditions the influence of the SCIR on ice stream flow. The very low basal traction along BIS allows the ice to be more responsive to perturbations in the stress regime in the coupled system. As a result, the SCIR slow the flow of BIS more than the flow of MacIS, even though MacIS is situated directly upstream of the SCIR (Figs. 9, S10). Similarly, differences in ice stream flow speeds with and without the SCIR extend further inland for BIS than for MacIS. A total speed change of $\geq 5$ ma$^{-1}$ extends 280 km upstream of the BIS grounding line, and 230 km upstream of the MacIS grounding line. This sticky spot distribution also has implications for mass flux (Table 3), the thickness gradient, and driving stress across the grounding line, as discussed in Section 4. BIS is more responsive in both the $\bar{B}_{inv}$ and $\bar{B}_u$ cases.

### 3.3.3 Longitudinal stresses $\bar{R}_{ll}$

The negative (compressive) and positive (tensile) components of longitudinal stresses $\bar{R}_{ll}$ are mapped and discussed separately for clarity. Both the SCIR and Roosevelt Island generate compression in the ice shelf between their locations and the grounding line (Fig. 6a and d). The pattern of compression differs between the $\bar{B}_{inv}$ and $\bar{B}_u$ cases. When a uniform rate factor is used, compression upstream of the SCIR is more diffuse and distributed over a larger area and longer section of the MacIS and BIS grounding line, while a spatially variable rate factor leads to more focused stress concentrations in some areas. Along the main trunk of MacIS, peaks in $-\bar{R}_{ll}$ form a 'rib-like' pattern characteristic of ice flow over sticky spots and an uneven subglacial bed topography (Fig. 6).

Differences in $-\bar{R}_{ll}$ between the reference and perturbed models are centred upstream of the SCIR and Roosevelt Island (Fig. 6c and f). Roosevelt Island is in the wake of the SCIR and therefore compression upstream of the island is larger when the SCIR are removed. Expressed another way, Roosevelt Island plays a larger role in generating compressive stresses when other obstacles make a lesser contribution. The reduction in compressive stresses with removal of the SCIR is limited to the MacIS grounding line in the $\bar{B}_{inv}$ case and includes both the MacIS and BIS grounding lines in the $\bar{B}_u$ case.

Where RIS flow is not impeded by pinning points, longitudinal tensile stresses vary between 50 and 150 kPa. In the reference model, $+\bar{R}_{ll}$ is largest ($>$200 kPa) where MacIS and BIS merge in the lee of the ice ridge separating the two streams. At this location, basal and lateral shear stresses make a lesser contribution to the force balance. The $+\bar{R}_{ll}$ pattern is stronger in the $\bar{B}_{inv}$ case (Fig. 7).

In general, flow obstructions such as pinning points act to reduce longitudinal tensile stresses upstream of their locations and the non-local nature of the momentum balance may allow this affect to extend upstream of the grounding line. In the spatially variable $\bar{B}_{inv}$ case, the SCIR act to decrease longitudinal tensile stresses near the ice stream grounding lines, while in the uniform rate factor $\bar{B}_u$ case, the SCIR have a lesser impact on tensile stresses upstream of the grounding line (Fig. 7c and f). Downstream of the SCIR, the difference between the reference and perturbed models is complicated, with a pattern that depends on both the prescription of $\bar{B}$ and on the geometry of the embayment.

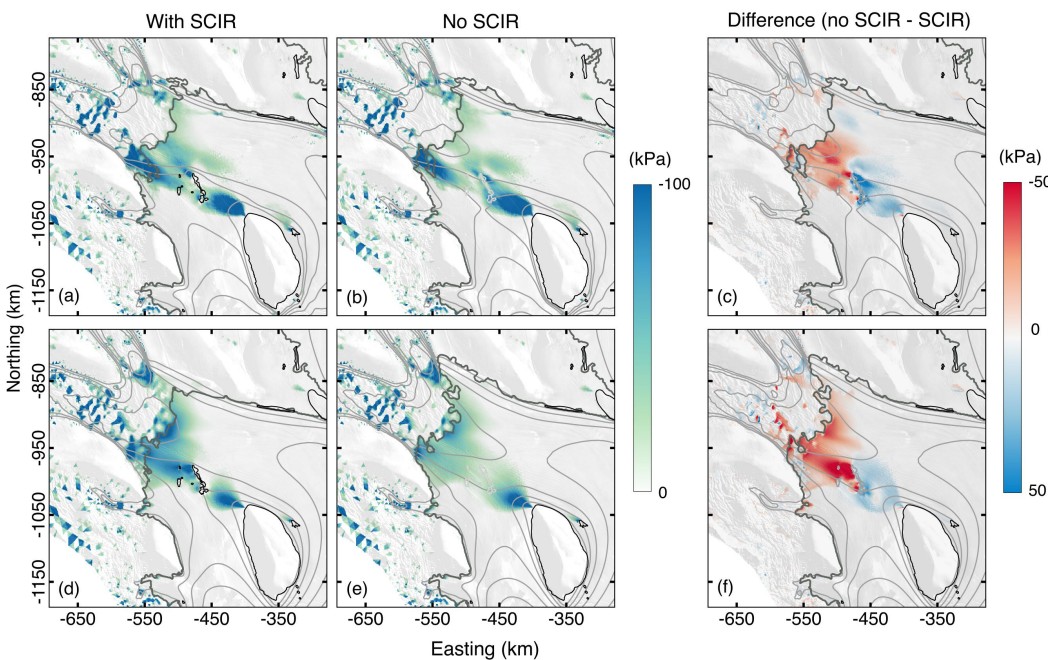

**Figure 6.** The longitudinal compressive stress $-\bar{R}_{ll}$ acting on the RIS and tributary ice streams with and without the SCIR. In (a-c), the simulation is initialised with $\bar{B}_{inv}$. In (d-f), the simulation is initialised with $\bar{B}_u$.

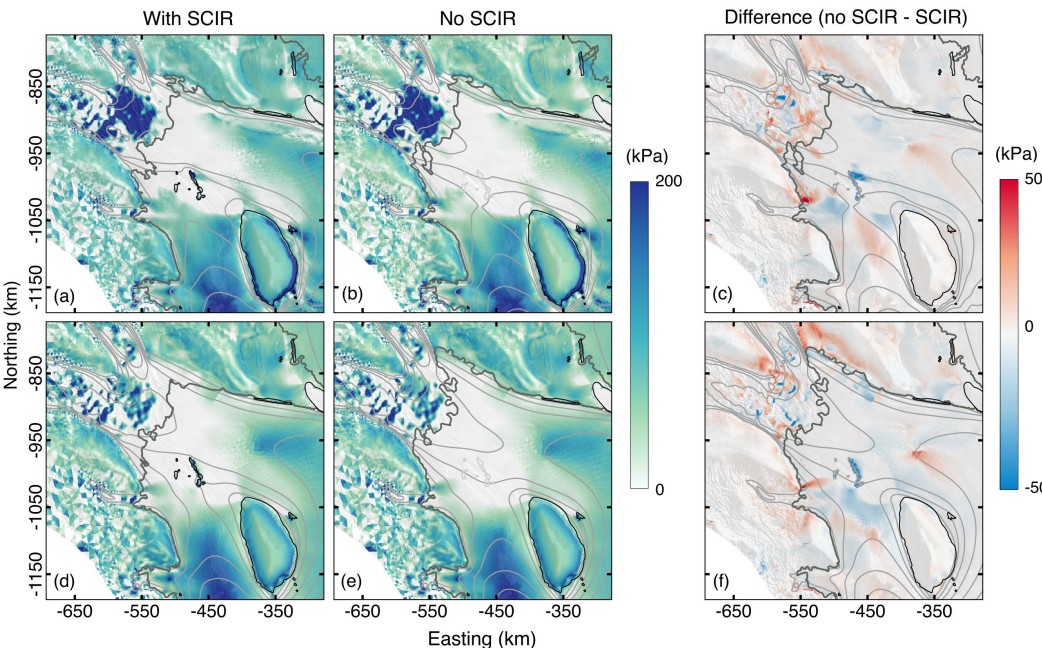

**Figure 7.** The longitudinal tensile stress $+\bar{R}_{ll}$ acting on the RIS and tributary ice streams with and without the SCIR. In (a-c), the simulation is initialised with $\bar{B}_{inv}$. In (d-f), the simulation is initialised with $\bar{B}_u$. In (a) and (b), the unusually high $+\bar{R}_{ll}$ values at the outlet of MacIS and BIS are due to relatively high $\bar{B}_{inv}$ values inferred during model initialisation.

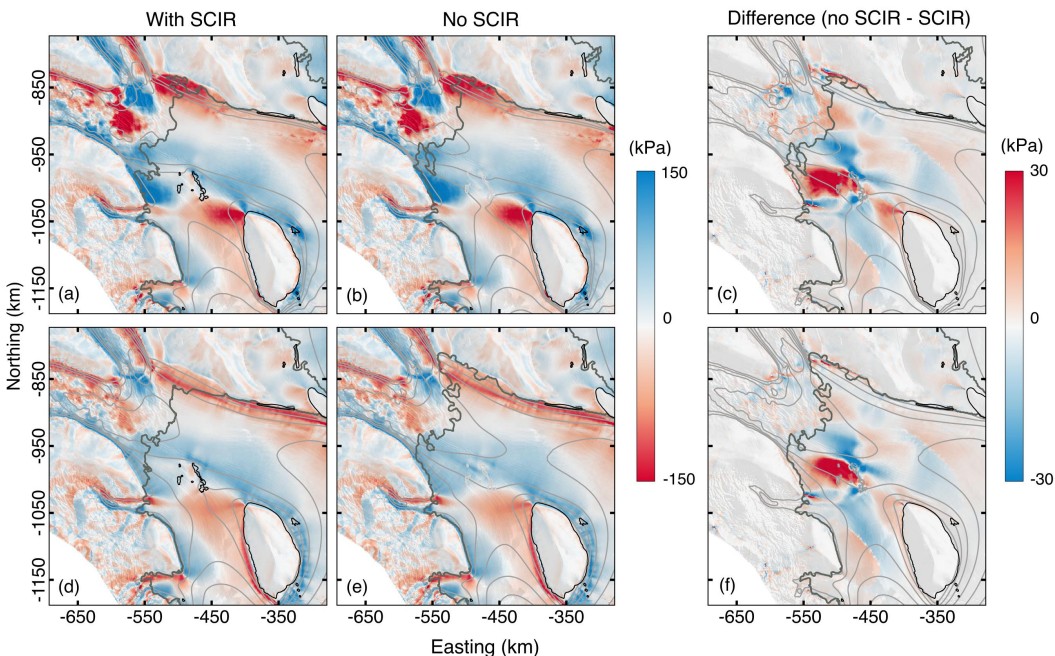

**Figure 8.** The lateral shear stress resisting RIS and tributary ice stream flow with and without the SCIR. Positive and negative $\bar{R}_{lt}$ magnitudes denote shearing along glacier-left or glacier-right margins, respectively. Difference maps show the absolute difference in $\bar{R}_{lt}$. In (a-c), the simulation is initialised with $\bar{B}_{inv}$. In (d-f), the simulation is initialised with $\bar{B}_u$. In (c) and (f), a positive (negative) change indicates an increase (decrease) in $\bar{R}_{lt}$ after removal of the SCIR.

### 3.3.4  Lateral shearing $\bar{R}_{lt}$

Ice flow past coastlines and pinning points generates lateral shear stresses $\bar{R}_{lt}$ (Fig. 8). Lateral shear stresses generated by the SCIR lead resistive stresses elsewhere to be smaller than would otherwise be required to balance the gravitational driving stress. In particular, $\bar{R}_{lt}$ is lower along the margins of Roosevelt Island and the Shirase Coast when the SCIR are present (positive differences in Fig. 8c and f). When the SCIR are removed, other coastal margins play a larger role in the stress balance via larger shear stresses. The pattern of re-partitioning between reference and perturbed models is broadly similar in both the $\bar{B}_{inv}$

and $\bar{B}_u$ cases (Fig. 8).

Together, the SCIR and Roosevelt Island support a band of relatively large shear strain rates (effectively, a shear margin) within the ice shelf. This margin is apparent as the large across-flow velocity gradient (Fig. 4a) and as right-lateral shear stresses south of the SCIR (Fig. 8). The deformation pattern is also apparent in the $\bar{B}_{inv}$ case as a band of relatively low ice stiffness (Fig. 2a). After the SCIR are removed, the magnitude of $\bar{R}_{lt}$ changes but the pattern persists (Fig. 8b and e). This persisting

band of high shear strain rates is expected for the $\bar{B}_{inv}$ case, where softer ice can continue to deform rapidly even after the SCIR 'boundary' is removed, but a similar outcome is also observed in the $\bar{B}_u$ case. Roosevelt Island thus appears to play an

important role in regulating flow from the ice streams and would act to divert mass flux southwestward even in the absence of the SCIR.

### 3.3.5 Transverse stresses $\bar{R}_{tt}$

Variations in transverse (across-flow) stress magnitudes are controlled primarily by the shape of the ice-shelf embayment. In a simple case without ice rises, transverse stresses are compressive and flow converges where embayment walls converge, while transverse stresses are tensile and flow diverges where embayment walls diverge. Ice rises and rumples modify the stress pattern by providing an additional lateral constraint.

The SCIR provide a lateral constraint by directing flow into two outlet pathways, between the ice rumples and the Shirase 310 Coast (to the glacier-right), and between the ice rumples and Siple Dome (to the glacier-left). In general, the SCIR reduce flow divergence in the region between Roosevelt Island, and the MacIS and BIS grounding line, with the pattern of $\bar{R}_{tt}$ depending on the selection of $\bar{B}_{inv}$ or $\bar{B}_u$ (Fig. S6). Localised increases in divergence originate from individual ice rumples in the SCIR complex as ice flows over each obstacle. The SCIR also increase convergence near the outlet of Echelmeyer Ice Stream (Fig. S7), but to the south, the SCIR create a diverging geometry and transverse tensile stresses that are locally larger 315 in comparison to the perturbed model without the ice rumples.

## 3.4 Ice-shelf flow, thickness and grounding-line position

The SCIR affect ice velocity across ∼30% of the RIS and their flow-regulating effect propagates far upstream of the MacIS, BIS and Echelmeyer Ice Stream grounding lines (Fig. 9c). The instantaneous change to velocity generated by removing the SCIR from the model domain (Figs. 9a and 10a) generates a change in ice flux, which in turn generates a time-dependent 320 adjustment that propagates as a feedback between ice thickness and velocity (Fig. 9). This feedback is relevant to interpretation of observed ice-shelf change.

When the SCIR are removed from the model domain, velocity magnitudes at their former location increase by up to 200 ma$^{-1}$ (100%) within 5 years. The mean instantaneous velocity change over the lower MacIS and BIS is, however, negative (Fig. 9a). Re-partitioning of resistive stresses in response to removal of the SCIR explains the instantaneous velocity decrease. 325 In particular, removal of the SCIR and the focused lateral shearing associated with the pinning points means that lateral shearing $\bar{R}_{lt}$ generated by other features is distributed over a wider region to satisfy the balance of forces. This redistribution has the effect, initially, of slowing flow south of the former SCIR, and slowing the flow speeds of MacIS and BIS. The magnitude of the basal friction parameter $\alpha$ moderates the ice-stream response (Section 3.3.5). Modification to the thickness field is required to overcome the change in the pattern of lateral shear stresses and by 5 years, velocity has increased throughout the region. 330 Following this, the rate of change decays as the velocity and thickness relax toward a new model steady-state.

The SCIR modify ice-shelf thickness by generating compression and thickening upstream, and extension and thinning downstream of their location (Fig. 11a). Following removal of the SCIR, ice immediately upstream of the former ice rumples thins, ice immediately downstream thickens and the surface elevation gradient diminishes to almost zero within 75 years (Fig. 10b). Adjustments in ice thickness extend upstream to MacIS and BIS, and downstream toward the calving front, although thickness

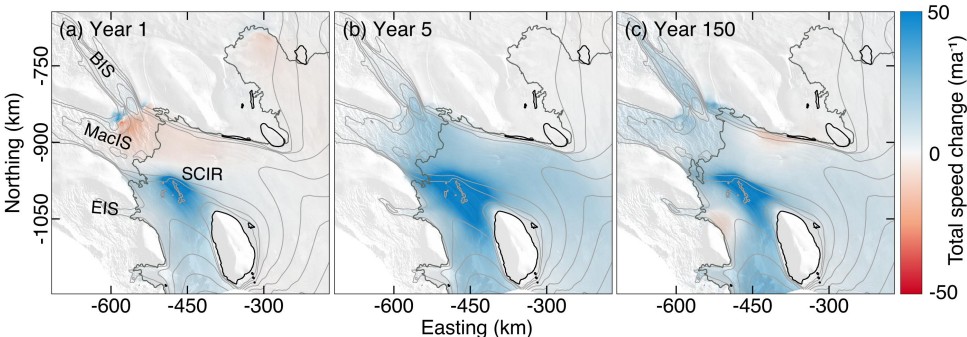

**Figure 9.** The total difference in ice speed between the steady-state, $\bar{B}_{inv}$ reference model (with SCIR) and the perturbed model (without SCIR) at various model timesteps following removal of the SCIR. (a) is the instantaneous response and (b-c) demonstrate the longer timescale adjustment of the ice-shelf and ice-stream system. Positive values indicate faster flow for the perturbed model without the SCIR and negative values indicate slower flow. By a timestep of 150 years, the model has reached a new steady-state. Grounding-line positions are from the perturbed model at timesteps of 1, 5 and 150 years. The velocity contour lines have an interval of 100 ma$^{-1}$.

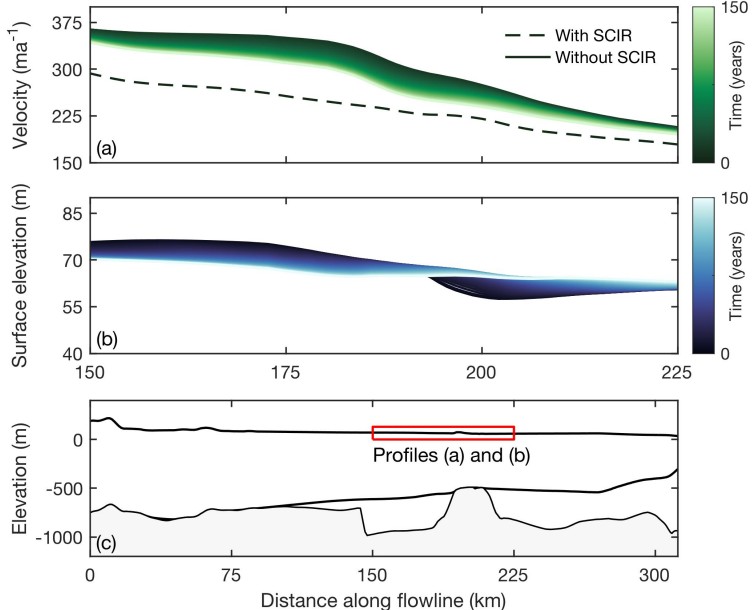

**Figure 10.** The adjustment in (a) ice velocity and (b) the along-flow surface gradient in response to removal of the SCIR (the $\bar{B}_{inv}$ case). The 'time' variable refers to the number of years after removal of the SCIR from the model domain. In (a), the difference between the dashed profile (flow speeds with the SCIR) and the profile at 0 years represents the instantaneous speed-up due to removal of the SCIR. (c) demonstrates the location of the profiles in (a) and (b). The location of (c) is indicated in Fig. 1b.

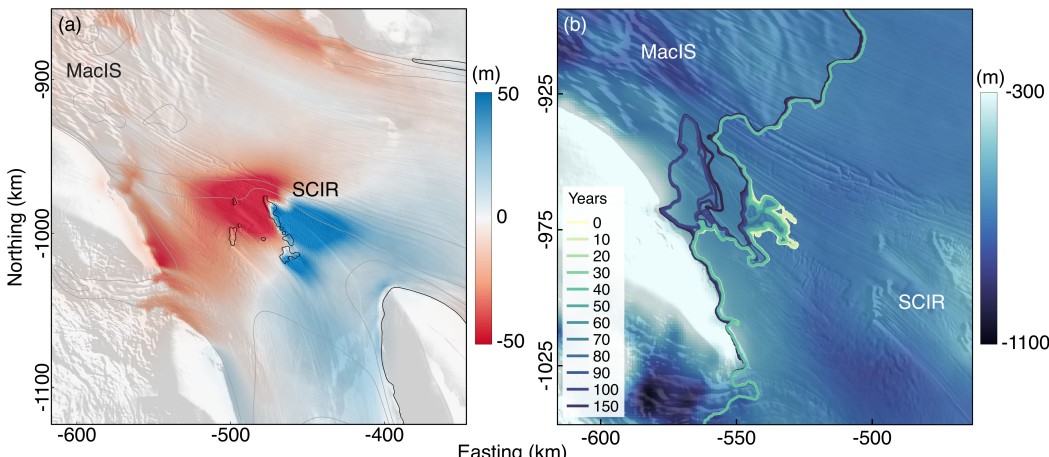

**Figure 11.** (a) The total change in ice thickness 150 years after removal of the SCIR. Red indicates thinner ice and blue indicates thicker ice without the ice rumples. (b) Retreat of a section of the simulated MacIS grounding line ($\bar{B}_{inv}$ case) following removal of the SCIR plotted on the subglacial bed elevation (Fretwell et al., 2013).

**Table 3.** Mass flux differences between the reference (with SCIR) and perturbed (no SCIR) models. The % increase is the total mass flux increase in response to removal of the SCIR. Flux gate locations are shown in Fig. 1b.

| Flux gate | Mass flux difference ($km^3 \ yr^{-1}$) | Mass flux % increase |
|---|---|---|
| a. MacAyeal Ice Stream | 0.496 | 2.37 |
| b. Bindschadler Ice Stream | 0.502 | 3.43 |
| c. Echelmeyer Ice Stream | 0.029 | 1.11 |
| d. Prestrud Inlet | 0.165 | 4.51 |
| e. Shirase Coast-Roosevelt Island gate | 1.199 | 6.39 |
| f. Roosevelt Island-Siple Dome gate | 1.141 | 4.33 |

change at the ice front is very small ($\sim$3 m). This implies that the fixed shelf front position is unlikely to have affected model experiment results. The magnitude and spatial pattern of the transient response is specific to the experimental design that intended to quantify the SCIR contribution to the mechanics of the RIS, rather than to investigate externally-forced change, such as pinning point modification due to basal melting. The fundamental mechanisms investigated here (i.e., the redistribution of stresses and the longer-timescale adjustment of ice flow and thickness), apply regardless of forcing.

Speed-up and thinning initiated by removal of the SCIR result in small adjustments to grounding-line position upstream of their former location. A 30-km-wide section of the MacIS grounding line retreats up to 40 km inland, where it merges with a large subglacial lake (Fig. 11b). Retreat stops when ice flow has adjusted to the perturbation and the grounding line reaches a rise in the subglacial topography (Fig. 11b).

The differences in velocity and thickness with and without the SCIR result in different mass flux patterns. The ice rumples

reduce flux across the grounding lines of nearby ice streams by several percent (Table 3). BIS is more responsive to the pinning points than MacIS due to the lower basal traction acting on BIS. Overall, total ice volume in the eastern RIS differs by only $\sim 1\%$ with and without the SCIR. Altogether, the pinning points cause a redistribution of mass (Figs. 10b and 11a) and only a small increase in the total ice volume stored in the eastern RIS.

## 4   Discussion

The magnitude of the flow resistance currently provided by the SCIR is of the same order of magnitude as the flow resistance provided by the larger and more well-grounded Roosevelt, Crary and Steershead Ice Rises (Still et al., 2019). This finding alone does not reveal how *important* the SCIR are to maintaining the present day configuration of the RIS. If stability is associated with grounding line position, then the simulations imply that the SCIR are unimportant (SCIR removal results in a transition to a new steady-state), despite the relatively large flow resistance they provide. This is due to a regional redistribution of ice

thickness and resistive stresses. The redistribution, which itself depends on embayment geometry, moderates the sensitivity of the coupled ice-sheet and ice-shelf system to the ice rumples. Similar redistributions should be expected for changes to other, individual pinning points. If stability is associated with crevasse and rift formation (Bassis and Ma, 2015; Borstad et al., 2017; Lai et al., 2020), the SCIR generate shear and tensile stresses that form crevasses and therefore the removal of pinning points may be expected to improve stability, although changes to shear stresses elsewhere may promote crevasse and rift formation in

those locations.

The model results suggest a connection between pinning points and grounded ice flow that involves basal traction, and thus basal properties, upstream of the grounding line. The SCIR increase the longitudinal compression acting on MacIS by 40 kPa (their influence on longitudinal stresses acting on BIS is negligible, Fig. 6c). The extra compression at the MacIS grounding line yields relatively thicker ice and thus a slightly larger driving stress in comparison to a configuration without the SCIR. In the

model, the initialised basal friction parameter reflects this larger driving stress. BIS, in contrast, is less obstructed and both the driving stress and basal traction are lower in comparison to MacIS. The lower basal traction explains the greater responsiveness of BIS to the SCIR in the model simulations. In the real system, such a coupling could manifest via stress-driven changes in basal water flow and till properties that in turn affect basal friction. A model in which basal till properties are coupled to the basal hydrologic system would be required to investigate this hypothesis further.

The difference between MacIS and BIS sensitivity to the SCIR highlights a connection between floating and grounded regimes that has been recently examined theoretically (Sergienko and Wingham, 2019) and deserves further investigation in the context of model initialisation. Following a related line of argument in their examination of the Thwaites Glacier (TG) response to loss of contact with the "Eastern Peak" pinning point, Nias et al. (2016) concluded that the TG system was more sensitive to selection of the basal friction parameter beneath the glacier than to unpinning. The present work is consistent

with their conclusion that the basal friction parameter assigned to upstream grounded ice conditions the response of the whole system, however, it does not lead to the conclusion that small-scale pinning points can be neglected from ice flow models.

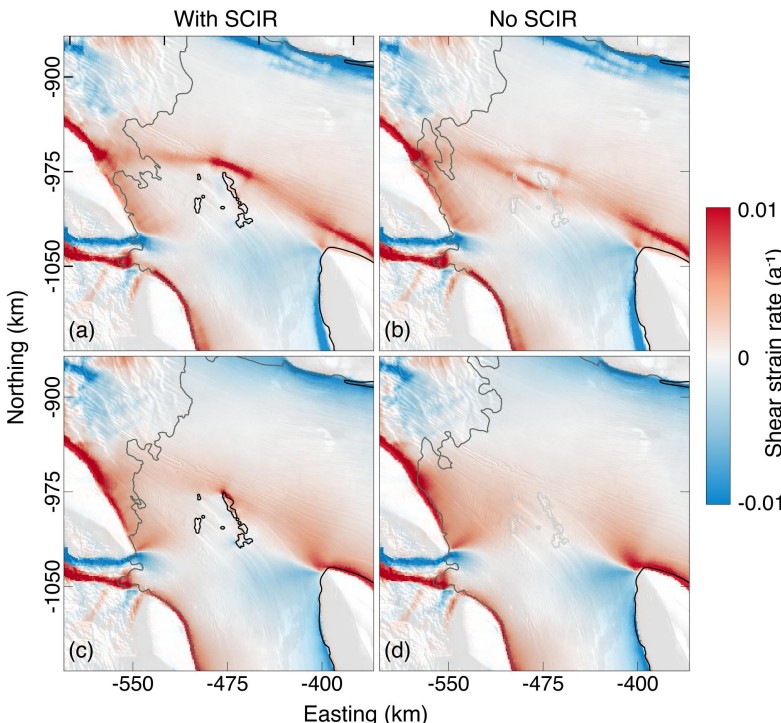

**Figure 12.** Model shear strain rates near the SCIR for (a) and (b) spatially variable ice properties ($\bar{B}_{inv}$ model) versus (c) and (d) uniform ice properties ($\bar{B}_u$ model). Grounding-line positions are from the appropriate $\bar{B}_{inv}$ or $\bar{B}_u$ reference and perturbed models.

Instead, the present work demonstrates how selection of the friction coefficient parameter for pinning point nodes during model initialisation modifies the flow of upstream grounded ice. This, as has been shown, emphasises the importance of the correct representation of pinning point morphology during initialisation.

The selection of $\bar{B}$ directly affects the pattern of resistive stresses and the simulated roles of different pinning points. When $\bar{B}$ is fitted to observed velocities, shear margins are represented as relatively narrow bands with lower $\bar{B}_{inv}$ (that is, ice with a greater readiness to deform under a given stress). In contrast, spatially uniform $\bar{B}_u$ distributes shearing over a wider across-flow distance (Fig. 12). Lower $\bar{B}_{inv}$ near the SCIR facilitates ice flow past the largest rumple in the complex and larger mass flux over and around the individual ice rumples. As a result, Roosevelt Island makes a larger contribution to balancing the gravitational driving stress in the $\bar{B}_{inv}$ case than the $\bar{B}_u$ case.

The importance of the correct representation of small–scale pinning points to model–inferred ice properties appears in several recent studies (Fürst et al., 2015; Favier et al., 2016; Berger et al., 2016). The experiment design used in the present work allows the mechanical and dynamical reasons for this sensitivity to be examined. Lower $\bar{B}$ within shear margins facilitates ice flow, limits upstream thickening and compression, and may, depending on pinning point location, condition the ice shelf to respond rapidly to unpinning. When pinning points are close to a coastal margin, as was the case in Favier et al. (2016), their effect is

to enhance shearing and reduce the inferred $\bar{B}$. Simulations in which the pinning points were included as part of their model initialisation experienced a larger magnitude response to external forcing (basal melting) than simulations in which the effect of small–scale pinning points on $\bar{B}$ was not included. From an ice dynamics point of view, this result is obtained at least in part because softer ice at the margin limits the transfer of resistive stresses generated by coastal features to the wider ice shelf.

Because $\bar{B}$ is a fixed property of model elements set during initialisation, this particular pinning-point effect persists after the pinning point is lost, and in turn, facilitates retreat. There may be unintended consequences of fixed, constant $\bar{B}$ as relatively low total strain can change crystallographic preferred orientation, and thus $\bar{B}$, within an ice shelf or ice stream (Duval and Castelnau, 1995; Lutz et al., 2020; Jordan et al., 2020). The SCIR also support a fast-deforming shear band (Fig. 12), but their mid-flow position means that the effect of the shear band after unpinning is limited in the present simulations.

The choice of basal friction law and its coefficients determine mass flux across the grounding line. The Budd-type friction law (Budd et al., 1979) used here is a common choice in ISSM applications (e.g., Seroussi et al., 2017; Haubner et al., 2018; Schlegel et al., 2018). In comparison to other possible friction laws, the Budd-type friction law is associated with a grounding line that is more sensitive to perturbations, and thus any changes in grounding-line position may be overestimated (Tsai et al., 2015; Brondex et al., 2017; Joughin et al., 2019). Different momentum equations, for example, full-Stokes instead of the SSA,

will also yield different results (Morlighem et al., 2010), particularly where thickness gradients are large, and this may explain the difference between our ISSM-inversion and force budget inferences of the basal friction parameter.

The inferred basal friction coefficient $\alpha$ encapsulates the mechanical and thermal properties of the ice/bed interface in a single, spatially varying, parameter. Individual physical processes that control basal sliding (e.g., till deformabililty, presence of subglacial meltwater, bedrock bumps) are therefore hidden within the friction coefficient pattern. Without additional pa-

rameterisations to account for change in these processes over time, $\alpha$ is held fixed. Given the apparent dynamical connections between pinning points and basal traction on grounded ice (identified here and by Nias et al. (2016)), improved representation of sliding and of the processes responsible for basal friction are likely to lead to new insights into the behaviour of the coupled system.

## 5 Conclusions

The flow-regulating effects of pinning points should be examined in the context of the interconnected ice-sheet and ice-shelf system. Despite their small size, the SCIR affect flow and thickness over ∼30% of the RIS, and the adjustment to ice-stream flow reaches >200 km upstream of the grounding line. When the SCIR are present, upstream compression and thickening steepens the along-flow surface gradient between the MacIS grounding line and the shelf front (Fig. 10). A steeper surface gradient increases the driving stress, which in turn maintains mass flux over and around obstacles in the flow field. Nonetheless,

the net effect of the SCIR is to slow ice flow and reduce mass flux. When the SCIR are removed, compression upstream of their location decreases, but upstream compression and lateral shearing around Roosevelt Island increase, and along-flow tensile stresses in the ice shelf remain largely unchanged. Altogether, the new balance of resistive stresses supports faster flow and increased mass flux, but with a negligible change in total ice volume.

Pinning points have been implicated as features that mediate the rate of grounding line retreat, yet their role in conditioning grounded ice flow has received less attention. In the present work, the direct effect of the SCIR on the momentum balance upstream of the grounding line is quantified and an indirect effect, via a connection involving ice stream basal traction, is suggested. This result is obtained by comparison of the relative sensitivity of the adjacent MacIS and BIS to the SCIR. Larger driving stresses acting on the grounded MacIS directly upstream of the pinning points require a relatively larger balancing basal drag, which in turn makes the grounded ice less sensitive to changes in pinning point configuration. These somewhat subtle effects emerge in the low driving stress, low basal traction environments of the Shirase Coast ice streams.

Pinning points contribute to the stress balance of the coupled ice-shelf and ice-sheet system, and therefore, any changes to pinning point configuration (formation, evolution or unpinning) will modify the system response to climate forcing and the rate of grounding line retreat. In the present case, the ice shelf remained in contact with other pinning features, allowing resistive stresses arising there to moderate system response to the SCIR. In a changing climate, those boundaries may also be affected, in which case their moderating effects may be limited. Because these interactions condition the grounding line response over multi-decadal time scales, processes related to pinning point formation, evolution and unpinning should be represented in numerical ice sheet models used to simulate the response of the ice-sheet and ice-shelf system to climate forcing.

As the climate continues to warm, ice shelves will continue to respond via changes in surface and basal mass balance. Modifications to ice shelf geometry will continue to drive changes in pinning point geometry, which will in turn drive changes in the mass and momentum balance in the ice-shelf and ice-sheet system. Improving the representation of pinning points requires attention to bathymetric data sets but also to the properties of subglacial material, either through realistic inferences of parameter values or better mathematical representations of the underlying physical processes. Although not addressed here, the incorrect representation of pinning points during model initialisation also has implications for the inference of the basal shear stress and ice softness parameter upstream of the grounding line. Naive inversion for the friction coefficient with no further manual tuning before model relaxation may lead to the incorrect representation of pinning point morphology, ice velocity, and the upstream flow resistance provided by pinning points. Over-simplifying or exaggerating the relative importance of pinning points in a modelling scenario may lead to over- and under-estimates of the role of pinning points in the ice-shelf and grounding-line response to climate forcing. The present contribution demonstrates the importance of high fidelity representation of pinning points for simulation of their effects in system models and, by extension, how observed change is interpreted. The use of spatially- and temporally-fixed ice properties and basal friction may also be insufficient to represent a changing pinning point environment. Prognostic simulations of pinning point dynamics will be further improved by the implementation of process-based ice properties and basal friction fields that respond to changing flow conditions.

*Data availability.* Model simulations used the open source Ice-sheet and Sea-level System Model (ISSM) available at https://issm.jpl.nasa.gov/ (last access: August 31, 2020, Larour et al. 2012). Datasets used to initialise the model are publicly available. Ice thickness and bathymetry are from the Bedmap2 compilation (Fretwell et al., 2013), available at https://secure.antarctica.ac.uk/data/bedmap2. MEaSUREs ice velocity

datasets (Rignot et al., 2011b) are available at https://nsidc.org/data/measures/data_summaries. Landsat 8 ice velocity datasets (Fahnestock et al., 2016) are available at https://nsidc.org/data/NSIDC-0733/versions/1.

*Author contributions.* Still conducted the numerical modelling. Still and Hulbe contributed equally to the analysis and writing.

*Competing interests.* The authors declare that they have no conflict of interest.

*Acknowledgements.* This work was supported by the New Zealand Post Antarctic Scholarship, the New Zealand Antarctic Research Institute (NZARI) funded Aotearoa New Zealand Ross Ice Shelf Programme, "Vulnerability of the Ross Ice Shelf in a Warming World", the New Zealand Antarctic Science Platform, and a University of Otago postgraduate publishing bursary. We thank the editor and two anonymous reviewers for their time and thoughtful comments, which helped us to improve the manuscript.

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
