# Peer review of "Mechanics and dynamics of pinning points on the Shirase Coast, West Antarctica"

_The Cryosphere, 2020_

## Referee Comment (RC1) · Anonymous Referee #1 · 17 Nov 2020

This paper provides a detailed modelling study of the effect that the Shirase Coast Ice Rumples have on the flow of the Ross Ice Shelf. The work looks technically sound and well carried out. The main issue I have with this paper is that it's a bit hard to see the forest for the trees. Specifically, there results are shown several ways and numerous figures are given that show various aspects of the with and without the rumples, but the discussion is fairly thin its not clear that it makes the case that ice rumples matter. Other than the fact the that the rumples will affect the velocity and the stress distributions in somewhat different ways with different assumptions in model parameters, it's not really clear to me whether they are essential features contributing to the overall stability of the ice shelf. Yes, the details of the shelf will differ depending on whether they are present, but does that really matter. I would think this manuscript would be much improved if

some kind of big picture statements could be made about the influence of the rumples on long-term stability (even if turns out the ice shelf behaves roughly the same, albeit with some changes in the details). This seems to be what Table 3 indicates (mass flux differences of only a few percent, which may still reflect transients from the abrupt and somewhat unphysical removal (actual ungrounding might be followed by regrounding due to the increased flux).

It might be good to remove a figure or table or 2 and put less emphasis on the all of the details, many of which have to do with the response to the abrupt removal of a bedrock feature, which not going to happen in reality. It's still a good thought experiment, so more discussion about the effect of the rumples on overall ice stability. How would the RIS be different in a parallel universe where the bathymetric features that give rise to the rumple did not form? Along these lines, is 150 years a long enough period to get rid of all transients associated with the step-removal of the topo. As noted above, the flux changes are relatively small, especially compared with everything else going on these ice streams over 150 year, so why do we care if our bottom line is sea level rise estimation. Especially if the this the difference between all or nothing. Maybe a good-enough pinning point representation is fine. I am not saying this is the case, rather I want to see the case made based on the overall system response, not just in the details presented here.

I think with a bit for refocussing and wordsmithing this could be an excellent paper.

Specific points Line 27 "and THE resulting..."

Line 59. Given we are getting much more frequent time series, it would be good soften the limited snap-show view statement, even if it may apply here. (At least go with snap-show viewS plural since we have very detailed time series for some areas).

Line 66. Hyphenate ice-sheet and ice-shelf, and separate with En dash (I think the En dash is currently there).

Equation 3 – Its disappointing to see this sliding law still being used. Why not use at least Weertman if not Coulomb plastic. At least add some discussion on the implications and validity of this this choice.

Line 102, Equation 4. Are you actually updating the model with this height above flotation effective pressure? It's not clear that this is appropriate for Siple Coast (or maybe anywhere) since most boreholes indicate effective pressures near zero.

Lines 118-125, please change all instances of resolution to posting – most of these data sets have nothing like this resolution (especially the bed products).

Paragraph beginning at line 131. It's unclear without looking at the supplement whether you are inverting for B on the floating ice and alpha on the grounded, or B on both. The text seems to indicate the latter, but Figure 2 seems to indicate the former since no value of B is shown on the grounded ice. Please clarify in main text. Along those lines, inversion for both B and alpha can be problematic. Yes, you will always get less model-data mismatch, but that's not necessarily a better fit since you have introduced an extra degree of freedom.

Line 149. A 2-year time step with speeds that potentially move ice 750 m (Fig. 1) through elements with 500-m dimension seems a bit dangerous with respect to CFL. Please justify.

Line 174: As a point that extends beyond this sentence, nothing is mentioned about whether any regularization has been applied in the inversions. Even if none has, that should be stated. Specific to this sentence, regularization might have smoothed out the friction coeff, so no manual adjustment would have been required.

Figure 3. With all of the map view figures in the main manuscript, nobody should have to skip to the supplement to see the locations of profiles. Please show in on or more figures in the main text.

Line 186. Remove the word "very" the numbers are similar, but given the uncertainties,

some of this may be due to chance, so just keep it at "similar".

Section 3.3 Another sentence or two here of introduction and why these equations are being given would be appropiate.

Table 1. How is their basal shear stress, however small, when the ice rumple is removed (where is the traction coming from). Maybe this is just an artifact of the force budget computation, but some kind of explanation is required (zero to within errors?).

Line 192. Minor point, but since 10b is referenced before 10a, they should be swapped, especially since there is no other compelling reason for the current ordering.

---

## Referee Comment (RC2) · Anonymous Referee #2 · 4 Dec 2020

In their study, Still & Hulbe analyze the mechanical impact of ice rumples off the Shirase Coast on the ice flow of Ross Ice Shelf and upstream. They use the ice flow model ISSM and initialize the model using optimization techniques to infer basal friction and ice softness for a regional setup of the Siple Coast and Ross Ice Shelf. The influence of the ice rumples is tested by first running the model into steady state with ice rumples present and then doing perturbation experiments. In the simulations, surface mass balance is based on Vaughan et al. (1999) and basal mass balance is based on a linear, depth-dependent parameterization like in Martin et al., 2011 with parameters adjusted to keep the grounding line position close to its present day location during the relaxation simulation. For the initial steady state the basal traction inferred for the ice rumples is adjusted so that the geometry of the rumples in steady state compares

with observations. Furthermore, a constant ice softness field is used instead of the inferred field to test the robustness of the results. In the perturbation experiments, the ice rumples are removed by digging them away in the topography, and the model is then integrated forward for 150 years and the response in driving stress, longitudinal stress, lateral stress, ice speed, ice thickness, grounding line position and shear strain rate is evaluated. This design of the perturbation experiments is a useful approach to test the effect of pinning points on ice dynamics. In addition, a force budget method is used on the SCIR and Roosevelt Island in the initial state and the perturbed state after 150 years.

The study is very detailed in the analysis of the stress changes, which makes it easy to lose track of the main results and conclusions and how they are reached. I think that a clearer argumentation and refocusing of the main findings would be very beneficial for the paper and make it an interesting contribution to The Cryosphere.

Major comments

- **Proposed feedback.** This comment concerns lines 9-10, 329-337 and 378-384. (1) I'm not sure I fully understand the proposed feedback: higher backstress from a pinning point is suggested to increase the ice thickness of the ice stream, thereby increasing the driving stress and basal drag which is then reflected by an increased occurence of sticky spots at the base of the ice stream, making the ice stream less responsive. But to close the loop, the last effect has to feed back on the backstress generated by the ice rumples. How does this work? (2) I don't understand how the proposed feedback between pinning points and basal traction of ice streams can be deduced from your experiments. In your invsersion you find higher basal friction coefficients in MacIS in comparison to BIS. But if those arise due to the presence of the SCIR cannot be singled out. It could also be that it is the local ice velocity together with the ice thickness field that determines the occurrence of sticky spots in the inverted basal friction

coefficient. This is not to say that it might not be possible, but I do not understand how the conclusion 'In the model, the larger basal drag acting on MacIS is itself, via regional changes in driving stress, a consequence of the coupled ice shelf and ice stream response to the SCIR.' can be drawn from the experiments presented in this study.

- **Basal friction adjustment and ice rise morphology.** This comment concerns lines 10-13, 345-346, 386-388 and Figure 3.
  (1) In the study, after the inversion procedure, basal friction coefficients of the ice rumples are adjusted in the relaxation simulations. I suppose that this is motivated by large-scale change in the ice rumple morphology when using the inverted basal friction coefficients in the relaxation runs? This would be interesting to extend on, and add the results of the relaxation simulation in Figure 3. Also it would be interesting to see how the overall results of this study would be affected by using the initially inverted basal friction fields.
  (2) Overall, a wrong morphology of the ice rumples after the relaxation simulations does not necessarily imply that the inversion produced wrong basal friction values as implies by statements in lines 10-12 and lines 368-388. It could also be that inconsistencies in the basal or surface mass balance or other factors causes a thinning, thickening, grounding or ungrounding of the ice rumples during the relaxation period. Don't get me wrong here, I think that your ad-hoc approach to correct the basal friction coefficient is ok. But I think that this should be discussed further and I'd be careful to blame the wrong morphology on the inverted friction coefficients alone. This should be extended on in the discussion.
  (3) Please be more clear: It is stated that 'by extension, any parameter that is affected by the initialization procedure' is represented incorrectly in lines 12-13. What parameters do you mean?
  Similarly, in line 345 it is stated that ' the present work demonstrates the role of pinning points in parameter selection during model initialization.' Please explain

more: What parameters are selected during the initialization procedure that are affected by the pinning points? Which role do the ice rises play? How is this shown in this study?

Furthermore, in line 389-390 is stated that 'The incorrect representation of pinning points also has implications for the inference of model parameters upstream of the grounding line during model initialization.' My understanding was that you did run the inversion to infer model parameters of basal shear stress and ice softness upstream of the grounding line based on the Bedmap2 geometry in which pinning points should be correctly represented. Or what do you refer to here?

- **Formulations**. Being more precise with your statements would make it easier for the reader to follow your ideas. For example in line 245 ('In general, the SCIR act to reduce longitudinal tensile stresses in grounded ice upstream of their location.'): in your next sentence you already mention that this is only partly true depending on the ice softness field used. Here you could directly go to the specific result or you should discuss why the general statement you made before (best supported by literature if it is not textbook knowledge) is actually not true in your experiments. See also comments on lines 293, 309, 319.

- **Figure captions**. Figure captions should give all relevant information on what is shown in the figure. For example, sometimes grounding lines are shown, but it is not indicated if this is an observed position or the position obtained in the relaxation simulation or in the respective experiment. In addition, the appropriate grounding lines should be displayed in figures that are interpreted to show changes at the grounding line or upstream (Figs 5,6,7,8,11,12). See also specific comments to the figures.

Further comments

- Line 9-10: see major comment.

- Line 10-13: see major comment.

- Line 15: 'transient' changes in ice shelf geometry in constrast to 'persistent' changes in ice streams. I'm not sure I understand this statement as the changes in ice thickness and speed that you present in Figures 9 and 11 are visible in both, the ice shelf and the ice streams.

- Figure 1: Add Echelmeyer Ice Stream as you are referring to it later on.

- Line 21: Another interesting study analyzing this is done by Pegler in 2018 ('Marine ice sheet dynamics: the impacts of ice-shelf buttressing').

- Lines 24 and 61: A bit of care with the wording should be taken here. The term 'flow-buttressing' has been used previously in Furst et al. It calculates the buttressing parameter by selecting the ice flow direction as a normal direction. However, the ice flow direction can be very different to the normal direction at the grounding line which is used in Gudmundsson 2013 to calculate a buttressing parameter.

- Lines 65 and 118: 'models' → 'model configurations' as ISSM is only one model?

- Section 2.2: How is the basal friction parameter set in regions that are not grounded during the inversion but that ground during the transient forward simulations? How is basal friction treated in elements along the grounding lines?

- Line 163: Why 150 years?

- Figure 2: It would be helpful to add here that also $B_u$ is shown in the colorbar of panel (a).

- Figure 3: It would be helpful to have (a) also the surface and velocity profile obtained with the inverted basal friction coefficient in the panels and (b) the magnitude of the optimized coefficient (e.g., averages along the lines). In addition, in panel (d) is the grey box showing the grounded regions in Bedmap2?

- Line 149: Is the same time stepping also applied in the perturbation experiments? If yes, how is the snapshot after 1 year shown in Figure 9(a) obtained?

- Line 175: How is the morphology for the different friction coefficients obtained? I suppose that the 1000years relaxation was run with different basal friction coefficients for the ice rumples?

- Line 184: How does the optimized value compare to this value (see also comment on Figure 3)?

- Line 192: Fig S2.

- Figure 4: Is this an instantaneous velocity difference or a difference obtained after running the relaxation for 1000years with the corresponding basal friction coefficient? What grounding line position is shown? If it is not shown here, it would be helpful to show the final grounding line positions after the relaxation runs to see how the ice rumple geometry is affected by the adjustment.

- Section 3.2: What is your main finding or conclusion from this comparison in relation with the later chapters? I think that it would help for the following chapters to analyze the difference and similarities between the results in light of the robustness of the results.

- Line 221-222: This statement seems to be true for the largest of the ice rumples but not for the smaller, second-largest one to the left?

- Line 222-226: Since the figure does not show a grounding line position, it is hard to say, but from a rough estimate it does not look like driving stress along the grounding lines of the glaciers main trunks change significantly? It would

be helpful if you (1) add the grounding lines in the Figure and (2) add the driving stress changes in Table 3. In addition, do you know what the blue spots in MacIS in the $B_u$ case are? Could they be numerical artifacts in individual mesh elements?

- Line 227: How do you conclude that changes in flow buttressing are equal in both cases?

- Line 235: Is this pattern consistent with the location of sticky spots and topographic features?

- Line 244: Not sure I understand this statement, Figure 8 shows particularly high lateral shear stress in this area?

- Line 244: 'exaggerated' → 'stronger'? Since $B_{inv}$ is obtained through inversion, I would expect the velocity and ice softness field to be closer to present-day than for the ad-hoc assumption of constant $B_u$. Thus, I would think of $B_{inv}$ as the reference simulation and $B_u$ as a test case to support robustness.

- Line 247: Fig. 6 → Fig. 7.

- Line 272: 'increase divergence downstream of their location' - this seems to depend on the ice softness and there is a large spot of decreased divergence (red) directly next to the rumples (on their western side) and downstream?

- Section 3.3.5: Maybe move this earlier so that you define 'sticky spots' before you discuss them in Section 3.3.2.

- Figure 9: which case is shown here, $B_{inv}$ or $B_u$?

- Line 285-286: It is really hard to tell from Figure 9 in which ice stream the speed increases more after 150 years. Would maybe be helpful to point to Table 3 here

and add also absolute and relative speed changes along the glaciers grounding lines (and maybe move figures to the SI). In addition, it would be interesting to have an estimate of how far speed changes extend inland for both glaciers.

- Section 3.4: I suggest to move this section before the changes in stresses are discussed to give the reader first an idea of how thickness, grounding line position and velocities change which then also makes it easier to interpret them with respect to changes in stresses.

- Line 293: Please be more precise here. How does this feedback work?

- Line 296-303: That you find an immediate slow-down upstream of the ice rumples is surprising and interesting to me. I think that your explanation that the initial slow will be reversed once the ice thins in the location of the ice rumples could be supported more: you could do an additional, simple experiment in which you do not only remove the ice rumples in the topography but also thin the ice at their former location so that the perturbed ice shelf is flatter (i.e., using the thickness distribution after 5 years) and then compare the instantaneous response. If the response is similar to your current $5$ year response, then the initial response can most likely be linked to the initial thickness distribution in your perturbation experiment.

- Line 309-311: Be more specific here, what do you mean with 'the fundamental mechanism are generic'?

- Line 310 'mechanics and dynamics' $\rightarrow$ 'mechanics of Ross Ice Shelf'?

- Line 319-320: Be more precise here. A redistribution of mass from where to where? Is it large or small? And how do the pinning points affect the efficiency?

- Discussion: Discussion should be extended to include also a discussion of the model choices done here (e.g., sliding law), potential drawbacks and limitations

of the methodology (e.g., assuming that present-day Ross Ice Shelf and the Siple Coast Ice Streams are in steady-state).

- Line 329-337: I'm not sure I understand this feedback, see also the main comment.

- Line 335-338: This sentence could be misunderstood to indicate that the studies of (van der Wel et al., 2013; Hoffman and Price, 2014) investigate a physical coupling between pinning points and ice stream basal properties (none of the studies includes dynamic ice shelves).

- Line 338: Looking into Table 3, the relative mass flux increases following SCIR removal of MacIS and BIS seem quite close when comparing it to other ice streams listed. I agree that it is interesting that BIS shows a similar and slightly higher response than MacIS which is located more directly upstream of the SCIR, but calling it a 'contrast' is maybe a bit too much.

- Lines 345-346: See major comment.

- Conclusions: in this section it would be great if you could put your findings into a broader context, e.g., discussing the vulnerability of the SCIR in a changing climate and the implications of your work in this context.

- Line 387-388: This could be misunderstood to mean that you did apply the feature-specific tuning during the inversion and not after the inversion. The second part of that sentence could be misinterpreted to state that the ice rumple morphology influences the overall results of this study, but this is not shown, as the results from Figs 5 to 12 are all done using the same basal friction coefficient for the ice rumples.

- Line 388-390: see major comment.

- Figure 10: Please also add the formerly grounded region in background of panels a and b.

- Fig S5: What is shown in the background of the figure?

---

## Author Comment (AC1) · 25 Jan 2021

**Response to Anonymous Referee #1**

Holly Still and Christina Hulbe

January 25, 2021

We would like to thank Referee #1 for their helpful feedback on our work. The reviewer's comments are in black and our responses are in blue. Excerpts from the improved manuscript are italicised.

**Comments by Reviewer #1**

This paper provides a detailed modelling study of the effect that the Shirase Coast Ice Rumples have on the flow of the Ross Ice Shelf. The work looks technically sound and well carried out. The main issue I have with this paper is that it's a bit hard to see the forest for the trees. Specifically, there results are shown several ways and numerous figures are given that show various aspects of the with and without the rumples, but the discussion is fairly thin its not clear that it makes the case that ice rumples matter. Other than the fact the that the rumples will affect the velocity and the stress distributions in somewhat different ways with different assumptions in model parameters, it's not really clear to me whether they are essential features contributing to the overall stability of the ice shelf. Yes, the details of the shelf will differ depending on whether they are present,but does that really matter. I would think this manuscript would be much improved if some kind of big picture statements could be made about the influence of the rumples on long-term stability (even if turns out the ice shelf behaves roughly the same, albeit with some changes in the details). This seems to be what Table 3 indicates (mass flux differences of only a few percent, which may still reflect transients from the abrupt and somewhat unphysical removal (actual ungrounding might be followed by regrounding due to the increased flux).

The magnitude of the flow resistance provided by the SCIR is quite large, given their low relief and relatively low basal traction. This was an important conclusion in Still et al. (2019). Here, we are concerned that the snap-shot like approach used in that paper (and by everybody else who investigates whether or not particular pinning points "matter") could be misleading. It could cause us to overestimate the importance of individual features because it does not allow for other compensating (re-balancing) effects. With these kinds of concerns in mind, we did not make any claim about the importance of these particular features. Is a 1% change in the volume of the RIS due to the presence of the SCIR large? We thought the possible interaction between pinning points (their effects on thickness and stresses) and driving stress and traction in grounded ice was interesting and would argue that this may be what is really important but as the reviewer notes, more work is required to investigate the connection.

We have added an additional paragraph to the discussion to include some big picture statements on the influence of the Shirase Coast Ice Rumples (SCIR) on long-term ice shelf stability:

*The magnitude of the flow resistance currently provided by the SCIR is of the same order of magnitude as the flow resistance provided by the larger and more well-grounded Roosevelt, Crary and Steershead Ice Rises (Still et al., 2019). This finding alone does not reveal how important the SCIR are to RIS stability. If an unstable ice shelf configuration is required for irreversible grounding line retreat (Weertman, 1974; Schoof, 2007), then the simulations imply that the SCIR are unimportant to stability (SCIR removal results in a transition to a new steady-state), despite the relatively large flow resistance they provide. This is due to a regional redistribution of ice thickness and resistive stresses. The redistribution, which itself depends on embayment geometry, moderates the sensitivity of the coupled ice sheet–ice shelf system to the ice rumples. Similar redistributions should be expected for changes to other, individual pinning points. If stability is associated with crevasse and rift formation (Bassis and Ma, 2015; Borstad et al., 2017; Lai et al., 2020), the SCIR generate shear and tensile stresses that form crevasses and therefore the removal of pinning points may be expected to improve stability, although changes to shear stresses elsewhere may promote crevasse and rift formation in those locations. If stability is associated with ice shelf thickness (Gudmundsson et al., 2019), the SCIR cause a regional redistribution of mass with a net effect of about a 1% change in ice shelf mass, implying very little impact on long-term stability.*

Mass flux differences in Table 3 do not reflect transients from pinning point removal - the system has reached a new steady-state without the ice rumples. We have improved line 160 to make this clearer: *The steady-state reference model is perturbed by excavating the bathymetry beneath the SCIR to prevent mechanical contact between the ice and seafloor, and stepped forward for 150 years with a timestep of one year. By 150 years, the rate of change in ice shelf volume is <0.001%, indicating that the model has reached a new steady-state.*

We agree that complete removal of the pinning points is unphysical. It is, however, the correct approach for quantifying the net effect of the SCIR on the momentum budget: we want to calculate the difference in resistive stresses, with and without the contribution of the ice rumples. We clearly did not explain this well enough in the original manuscript and have tried to improve it in several ways:

Abstract, lines 1-5: *Ice rises and rumples, sites of localised ice-shelf grounding, modify ice shelf flow by generating lateral and basal shear stresses, upstream compression and downstream tension. Studies of pinning points typically quantify this role indirectly, through related metrics such as a buttressing number. Here, we quantify the dynamic effects of pinning points directly, by comparing model-simulated stress states in the Ross Ice Shelf (RIS) with and without a specific set of pinning points located downstream of the MacAyeal and Bindschadler Ice Streams (MacIS and BIS, respectively).*

Introduction, line 44: *Altogether, the ice shelf is a coupled system in which a change in any specific location may, though the momentum and mass balances, drive change in resistive stresses and ice thickness elsewhere. The aim of the present work is to quantify the complete system of mass and momentum adjustments caused by a specific set of pinning points in the RIS, West Antarctica.*

Introduction, line 63: *We take a different approach to examine the dynamical role of pinning points. Our aim is to quantify the complete pinning point contribution to ice dynamics, and we do this by performing numerical model simulations of ice-shelf and ice-stream flow with and without a collection of lightly grounded,*

*low relief ice rumples (hereafter called the Shirase Coast Ice Rumples, SCIR). Differences between two steady states – with and without the SCIR – show how the coupled system responds to their presence, including a repartitioning of resistive stresses and a redistribution of ice mass.*

It might be good to remove a figure or table or 2 and put less emphasis on the all of the details, many of which have to do with the response to the abrupt removal of a bedrock feature, which not going to happen in reality. It's still a good thought experiment, so more discussion about the effect of the rumples on overall ice stability. How would the RIS be different in a parallel universe where the bathymetric features that give rise to the rumple did not form?

The purpose of the model experiments is to "quantify the specific contribution [of the SCIR] to flow dynamics." There are two ways to do this, (1) a force budget analysis and (2) removing the features from a calibrated model, finding a new steady state, and subtracting one from the other. A third, related, approach is to compute a buttressing number. We've done all three and the results are not identical.

We would like to be clear that the model has not been used to conduct a thought experiment. It has been used to isolate specific terms in a non-local force balance. It may be that using the word *experiment* to describe the calculations muddles this and we have replaced it with *simulation* wherever this seemed to help. It is also possible that the word *removal* is itself unhelpful to our cause because it sounds like the "parallel universe" we did not set out to create.

We have refocused the abstract and introduction section to state more emphatically that we are quantifying the specific contribution of the Shirase Coast Ice Rumples to the flow dynamics of the Ross Ice Shelf and tributary ice streams, rather than conducting a 'thought experiment':

Abstract, line 2: *Here, we quantify the dynamic effects of pinning points directly, by comparing model-simulated stress states in the Ross Ice Shelf (RIS) with and without a specific set of pinning points located downstream of the MacAyeal and Bindschadler Ice Streams (MacIS and BIS, respectively).*

Introduction, line 64: *Our aim is to quantify the complete dynamical effect of a set of pinning points and we do this using numerical model simulations of ice shelf and ice stream flow with and without a collection of lightly grounded, low relief ice rumples downstream of the Shirase Coast (hereafter called the Shirase Coast Ice Rumples, SCIR). Differences between two steady states – one with and one without the SCIR – show how the coupled system responds to their presence, including repartitioning of resistive stresses and redistribution of ice mass.*

Please see our response to the previous comment for our statements on the influence of the Shirase Coast Ice Rumples (SCIR) on overall ice shelf stability.

Along these lines, is 150 years a long enough period to get rid of all transients associated with the step-removal of the topo.

Yes, the rates of change in model ice volume, thickness, and speed were tracked to identify when a ∼steady state had been achieved. We have clarified this in the text (line 160):

*The steady-state reference model is perturbed by excavating the bathymetry beneath the SCIR to prevent mechanical contact between the ice and seafloor, and stepped forward for 150 years with a timestep of 1 year.*

*By 150 years, the rate of change in ice shelf volume is <0.001%, indicating that the model has reached a new steady-state.*

As noted above, the flux changes are relatively small, especially compared with everything else going on these ice streams over 150 year, so why do we care if our bottom line is sea level rise estimation. Especially if the this the difference between all or nothing. Maybe a good-enough pinning point representation is fine. I am not saying this is the case, rather I want to see the case made based on the overall system response, not just in the details presented here.

We do present the overall system response in the manuscript, with a focus on the connection between the pinning points and the grounded ice streams. If the end-goal is sea level rise estimation, then the presence of the Shirase Coast Ice Rumples alone will have a negligible impact on the contribution to sea level rise. This is in part due to the redistribution of stresses and mass that the study aimed to examine. However, there are quite a few ice rises and rumples embedded in Antarctic ice shelves, more than 500 (Matsuoka et al., 2015; Moholdt and Matsuoka, 2015), and collectively these features will have a significant impact on mass flux into Antarctic ice shelves.

I think with a bit for refocussing and wordsmithing this could be an excellent paper

Thank you. We have revised the abstract, introduction and discussion sections to, we hope, be more clear about the rationale for the experiment design and connections with ice shelf stability. These changes are noted throughout our responses to both Reviewer 1 and Reviewer 2.

Line 59. Given we are getting much more frequent time series, it would be good soften the limited snap-show view statement, even if it may apply here. (At least go with snap-show viewS plural since we have very detailed time series for some areas).

Changed as suggested.

Line 66. Hyphenate ice-sheet and ice-shelf, and separate with En dash (I think the Endash is currently there).

We have added hyphens as suggested, although we prefer 'ice-sheet and ice-shelf system' rather than 'ice-sheet–ice-shelf system'.

Equation 3 – Its disappointing to see this sliding law still being used. Why not use at least Weertman if not Coulomb plastic. At least add some discussion on the implications and validity of this this choice.

We used a linear Budd-type friction law (Budd et al., 1979). This friction law is extensively used by the ISSM community (Bondzio et al., 2016; Seroussi et al., 2017; Haubner et al., 2018; Schlegel et al., 2018; Smith-Johnsen et al., 2020). The Coulomb plastic friction law may be more appropriate for application to ice streams, however this is currently not implemented in ISSM. The Weertman friction law (Weertman, 1957) intends to describe ice motion over hard bedrock and therefore is less appropriate for the fast-flowing Ross ice streams that lie over soft subglacial till with water present. In addition, the Weertman friction law has been tested in ISSM by Seroussi and Morlighem (2018) and they stated *"This friction law induces a sharp discontinuity in basal friction at the grounding line that is not realistic and not appropriate for problems investigating grounding line evolution but remains nevertheless widely used in the community (Brondex et al., 2017)"*.

During model tuning we tested different values of the friction law exponents $p$ and $q$. As a general rule, $(p, q) = (3,1)$ is appropriate for high normal stresses ($> 500$ kPa) (Budd et al., 1979) while $(p, q) = (1,1)$ is commonly used in ice sheet models to improve model stability (e.g., Morlighem et al., 2010; Bondzio et al., 2016). This is one reason why we used $(p, q) = (1,1)$. We also found that $(p, q) = (1,1)$ provided a more realistic representation of ice flow near the grounding line for our model of the Ross Ice Shelf and tributary ice streams than $(p, q) = (3,1)$.

We have added two paragraphs into the discussion section that consider the implications of our choice of friction law:

*The choice of friction law and its coefficients determine ice stream speeds and mass flux across the grounding line. The Budd-type friction law (Budd et al., 1979) used here is a common choice in ISSM applications, where it has been found to improve model stability and representation of grounding line migration (Seroussi and Morlighem, 2018). The grounding line is more sensitive to change than would be the case for other possible friction laws (Tsai et al., 2015; Brondex et al., 2017; Joughin et al., 2019).*

*The inferred basal friction coefficient $\alpha$ encapsulates the mechanical and thermal properties of the ice/bed interface in a single, spatially varying, parameter. Individual physical processes that control basal sliding (e.g., till deformabililty, presence of subglacial meltwater, bedrock bumps) are therefore hidden within the friction coefficient distribution. Without additional parameterisations to account for change in these processes over time, $\alpha$ is held fixed. Given the apparent dynamical connections between pinning points and basal traction on grounded ice (identified here and by Nias et al. (2016)), improved representation of sliding and of the processes responsible for basal friction are likely to lead to new insights into the behaviour of the coupled system.*

Line 102, Equation 4. Are you actually updating the model with this height above flotation effective pressure? It's not clear that this is appropriate for Siple Coast (or maybe anywhere) since most boreholes indicate effective pressures near zero.

The effective pressure $N$ is updated at each time step as the ice thickness changes.

Lines 118-125, please change all instances of resolution to posting - most of these data sets have nothing like this resolution (especially the bed products).

We have modified the sentence about the ice velocity datasets to clarify that the **grid** resolution of the Landsat 8 and Measures velocity datasets is 750 m and 900 m, respectively.

In the Supplementary Material lines 7-9, we stated that: *"The Bedmap2 sub-ice shelf bathymetry used to create the model geometry is interpolated from limited resolution (55 km) point measurements onto a 1 km grid (Fretwell et al., 2013). Predictably, the depth of the water column beneath the RIS is incorrect in some areas and this poses particular problems where the seafloor is unrealistically shallow."* This provides the justification for modifying the Bedmap2 bathymetry in the model geometry to ensure that the Ross Ice Shelf pinning points have the correct area of grounding.

Paragraph beginning at line 131. It's unclear without looking at the supplement whether you are inverting for B on the floating ice and alpha on the grounded, or B on both. The text seems to indicate the latter, but Figure 2 seems to indicate the former since no value of B is shown on the grounded ice. Please clarify in

main text. Along those lines, inversion for both B and alpha can be problematic. Yes, you will always get less model-data mismatch, but that's not necessarily a better fit since you have introduced an extra degree of freedom.

We inverted for $\bar{B}$ on floating and grounded ice. In Figure 2, we included $\bar{B}$ for floating ice only to make the point that the velocity of the floating ice shelf is primarily controlled by $\bar{B}$ while the velocity of grounded ice is primarily controlled by the inferred friction coefficient. We used two separate inversions to infer $\bar{B}$ and $\alpha$ and did not invert for both parameters at the same time.

We have modified this part of the method, line 133, as follows:

1. ***First inversion for*** $\bar{B}$ ***for floating and grounded ice***. *The initial estimate is a uniform* $\bar{B}$ *of* $1.6 \times 10^8$ *Pa s$^{1/3}$, corresponding to an ice temperature of -16.7°C...*

Line 149. A 2-year time step with speeds that potentially move ice 750 m (Fig. 1) through elements with 500-m dimension seems a bit dangerous with respect to CFL. Please justify.

The mesh was manually refined near pinning point perimeters to achieve a finer mesh resolution. Using the mesh generation algorithm implemented in ISSM, a **minimum edge length of 500 m** was assigned. In combination with anisotropic mesh refinement according to the distribution of velocity gradients, this resulted in a mesh resolution of approximately 1 km over the Shirase Coast Ice Rumples (ice velocity = 200 m/a), and a mesh resolution of approximately 3 km in the fast-flowing narrow of Bindschadler Ice Stream where ice velocity exceeds 600 m/a. The central Ross Ice Shelf, where velocities exceed 750 m/a has the coarsest mesh resolution (up to 10 km). The simulations were stable and we satisfied the CFL condition.

It was incorrect to state that the mesh resolution was 500 m when this was actually the minimum possible edge length specified when generating the mesh. In the manuscript (Line 81), we have rewritten this as:

*Mesh resolution ranges from approximately 1 km over pinning points, ice stream shear margins and the grounding line, to approximately 10 km over the inland ice sheet and central ice shelf.*

Line 174: As a point that extends beyond this sentence, nothing is mentioned about whether any regularization has been applied in the inversions. Even if none has, that should be stated. Specific to this sentence, regularization might have smoothed out the friction coeff, so no manual adjustment would have been required.

We applied Tikhonov regularisation. Rather than performing an L-curve analysis, the weighting was chosen such that we were able to capture the flow velocity gradients generated by the smallest ice rumples, but with an appropriate level of regularisation to avoid overfitting to noise in the velocity data (i.e., we reproduced the velocity gradients near the Shirase Coast Ice Rumples but have not reproduced any of the artefacts in the MEaSUREs velocity dataset, or the discontinuity between the MEaSUREs and Landsat 8 velocity datasets. The goal was to use the minimum amount of smoothing required to resolve velocity gradients generated by the smallest ice rumples in the domain.

A higher weighting of the Tikhonov regularisation cost function may have smoothed the friction coefficient over all of the ice rumples in the SCIR complex, however, the degree of regularisation required to smooth out the friction coefficient over the ice rumples (a 200 km$^2$ area) would result in a loss of spatial detail elsewhere

in the model domain. For example, the individual sticky spots along MacIS and BIS would not be resolved, and neither would the different density of sticky spots along MacIS and BIS.

We have added the following information to 2.2 Model initialisation: *The inverse method seeks to minimise the value of a cost function, a measure of the misfit between observed and modelled ice velocities, integrated over the whole model domain. Terms in the cost function include absolute and logarithmic misfits, and a third regularisation term is included to prevent physically unrealistic variations in $\bar{B}$ and $\alpha$ over short spatial scales.*

Figure 3. With all of the map view figures in the main manuscript, nobody should have to skip to the supplement to see the locations of profiles. Please show in on or more figures in the main text.

We have moved the profile locations and flux gate locations from the supplement to the first figure in the main manuscript. The improved figure is included at the end of this document (Fig. 1).

Line 186. Remove the word 'very' the numbers are similar, but given the uncertainties, some of this may be due to chance, so just keep it at 'similar'.

Changed as suggested.

Section 3.3 Another sentence or two here of introduction and why these equations are being given would be appropriate.

We have added the following statements to introduce the resistive stress calculations at the beginning of Section 3.3. *The non-local nature of the ice shelf momentum balance is examined in detail by computing the distribution of resistive stresses acting on RIS flow. The gravitational driving stress $\tau_d$ must be balanced by resistive stresses including the longitudinal stress $\bar{R}_{ll}$, transverse stress $\bar{R}_{lt}$, and the lateral shear stress $\bar{R}_{tt}$. Resistive stresses are computed using flow-following longitudinal $\dot{\epsilon}_{ll}$, transverse $\dot{\epsilon}_{tt}$ and shear $\dot{\epsilon}_{lt}$ strain rates from the model via Glen's flow law...*

Table 1. How is their basal shear stress, however small, when the ice rumple is removed (where is the traction coming from). Maybe this is just an artifact of the force budget computation, but some kind of explanation is required (zero to within errors?).

For floating ice without the ice rumples, basal drag should equal zero or fall within the uncertainty range of zero given the model-derived datasets used for the force budget calculations. Uncertainty in the force budget calculation arises from: (1) the errors in the modelled velocity data and strain rates; (3) the modelled ice thickness; and (3) the inferred inverse rate parameter $\bar{B}$.

This is presented in detail in Still et al. (2019) and referenced in the manuscript.

Although error propagation is not applicable to model-generated data (because there are no "measurement" errors to propagate), $\mathbf{F}_e$ (and therefore basal drag) is likely to fall within the uncertainty range of 0 given the large uncertainty bounds around $\mathbf{F}_e$ computed from observational datasets for Crary Ice Rise, Steershead Ice Rise, the SCIR and Roosevelt Island).

Line 192. Minor point, but since 10b is referenced before 10a, they should be swapped, especially since there is no other compelling reason for the current ordering.

We have switched the order in the revised manuscript. Please see Fig. 2 (in this document) for the updated figure.

**Updated figures**

[Figure]

Figure 1: Pinning points in the RIS and the model domain boundary. In panel (a), large pinning points are labelled: SCIR = the Shirase Coast Ice Rumples, RI = Roosevelt Island, SIR = Steershead Ice Rise and CIR = Crary Ice Rise. The colour map of surface ice velocity magnitude is from the MEaSURES velocity dataset (Rignot et al., 2011). The black line indicates the grounding zone (Bindschadler et al., 2011). Panel (b) shows the along-flow cross-sections intersecting the SCIR in Figs. 3 and 10 and the gates used for mass flux calculations in Table 3. The colour map of ice thickness is from the Bedmap2 compilation (Fretwell et al., 2013). In each figure from hereon, datasets are mapped with a Polar Stereographic Projection with a central meridian of 0° and a standard latitude of 71°S, and in most cases, overlayed onto the MODIS MOA (Haran et al., 2014).

[Figure]

Figure 2: The adjustment in (a) the along-flow surface gradient and (b) ice velocity in response to removal of the SCIR. The 'time' variable refers to the number of years after removal of the SCIR from the model domain. In (b), the difference between the dashed profile (flow speeds with the SCIR) and the profile at 0 years represents the instantaneous speed-up due to removal of the SCIR. (c) demonstrates the location of the profiles in (a) and (b). The location of (c) is indicated in Fig. 1b.

**References**

Bassis, J. N. and Ma, Y (2015). Evolution of basal crevasses links ice shelf stability to ocean forcing. *Earth and Planetary Science Letters*, **409**, pp. 203–211. DOI: 10.1016/j.epsl.2014.11.003.

Bindschadler, R, Choi, H, Wichlacz, A, Bingham, R, Bohlander, J, Brunt, K, Corr, H, Drews, R, Fricker, H, Hall, M, Hindmarsh, R, and Kohler, J (2011). Getting around Antarctica: new high-resolution mappings of the grounded and freely-floating boundaries of the Antarctic ice sheet created for the International Polar Year. *The Cryosphere*, **5** (3), pp. 569–588. DOI: 10.5194/tc-5-569-2011.

Bondzio, J. H., Seroussi, H., Morlighem, M., Kleiner, T., Rückamp, M., Humbert, A., and Larour, E. Y. (2016). Modelling calving front dynamics using a level-set method: application to Jakobshavn Isbrae, West Greenland. *The Cryosphere*, **10**, pp. 497–510. DOI: 10.5194/tc-10-497-2016.

Borstad, C., Mcgrath, D., and Pope, A. (2017). Fracture propagation and stability of ice shelves governed by ice shelf heterogeneity. *Geophysical Research Letters*, **44**, pp. 4186–4194. DOI: 10.1002/2017GL072648.

Brondex, J., Gagliardini, O., Gillet-Chaulet, F., and Durand, G. (2017). Sensitivity of grounding line dynamics to the choice of the friction law. *Journal of Glaciology*, **63** (241), pp. 854–866. ISSN: 00221430. DOI: 10.1017/jog.2017.51.

Budd, W. F., Keage, P. L., and Blundy, N. A. (1979). Empirical Studies of Ice Sliding. *Journal of Glaciology*, **23** (89), pp. 157–170. DOI: 10.1017/S0022143000029804.

Fretwell, P. et al. (2013). Bedmap2: improved ice bed, surface and thickness datasets for Antarctica. *The Cryosphere*, **7** (1), pp. 375–393. DOI: 10.5194/tc-7-375-2013.

Gudmundsson, G. H., Paolo, F. S., Adusumilli, S., and Fricker, H. A. (2019). Instantaneous Antarctic ice sheet mass loss driven by thinning ice shelves. *Geophysical Research Letters*, **46** (23), pp. 13903–13909. ISSN: 0094-8276. DOI: 10.1029/2019GL085027.

Haran, T., Bohlander, J., Scambos, T., Painter, T., and Fahnestock, M. (2014). MODIS Mosaic of Antarctica 2008-2009 (MOA2009) Image Map. DOI: 10.7265/N5KP8037.

Haubner, K., Box, J. E., Schlegel, N. J., Larour, E. Y., Morlighem, M., Solgaard, A. M., Kjeldsen, K. K., Larsen, S. H., Rignot, E., Dupont, T. K., and Kjaer, K. H. (2018). Simulating ice thickness and velocity evolution of Upernavik Isstrøm 1849-2012 by forcing prescribed terminus positions in ISSM. *The Cryosphere*, **12**, pp. 1511–1522. DOI: 10.5194/tc-12-1511-2018.

Joughin, I., Smith, B. E., and Schoof, C. G. (2019). Regularized Coulomb Friction Laws for Ice Sheet Sliding: Application to Pine Island Glacier, Antarctica. *Geophysical Research Letters*, **46** (9), pp. 4764–4771. ISSN: 0094-8276. DOI: 10.1029/2019GL082526.

Lai, C.-Y., Kingslake, J., Wearing, M., Po-Hsuan Chen Cameron, Gentine, P., Li, H., Spergel, J., and Wessem, J. M. van (2020). Vulnerability of Antarctica's ice shelves to meltwater-driven fracture. *Nature*, **584**, pp. 574–578. DOI: 10.1038/s41586-020-2627-8.

Matsuoka, K., Hindmarsh, R. C. A., Moholdt, G., Bentley, M. J., Pritchard, H. D., Brown, J., Conway, H., Drews, R., Durand, G., Goldberg, D., Hattermann, T., Kingslake, J., Lenaerts, J. T. M., Martín, C., Mulvaney, R., Nicholls, K. W., Pattyn, F., Ross, N., Scambos, T., and Whitehouse, P. L. (2015). Antarctic ice rises and rumples: Their properties and significance for ice-sheet dynamics and evolution. *Earth Science Reviews*, **150**, pp. 724–745. DOI: 10.1016/j.earscirev.2015.09.004.

Moholdt, G. and Matsuoka, K. (2015). Inventory of Antarctic ice rises and rumples (version 5) [Data set]. DOI: 10.21334/npolar.2015.9174e644.

Morlighem, M., Rignot, E., Seroussi, H., Larour, E., Ben Dhia, H., and Aubry, D. (2010). Spatial patterns of basal drag inferred using control methods from a full-Stokes and simpler models for Pine Island Glacier, West Antarctica. *Geophysical Research Letters*, **37** (14), p. L14502. DOI: 10.1029/2010GL043853.

Nias, I. J., Cornford, S. L., and Payne, A. J. (2016). Contrasting the modelled sensitivity of the Amundsen Sea Embayment ice streams. *Journal of Glaciology*, **62** (233), pp. 552–562. DOI: 10.1017/jog.2016.40.

Rignot, E, Mouginot, J, and Scheuchl, B (2011). MEaSUREs InSAR-Based Antarctica Ice Velocity Map. DOI: 10.5067/measures/cryosphere/nsidc-0484.001.

Schlegel, N., Seroussi, H., Schodlok, M. P., Larour, E. Y., Boening, C., Limonadi, D., Watkins, M. M., Morlighem, M., and Van Den Broeke, M. R. (2018). Exploration of Antarctic Ice Sheet 100-year contribution to sea level rise and associated model uncertainties using the ISSM framework. *The Cryosphere*, (12), pp. 3511–3534. DOI: 10.5194/tc-2018-105.

Schoof, C. (2007). Ice sheet grounding line dynamics: Steady states, stability, and hysteresis. *Journal of Geophysical Research*, **112** (F3), F03S28. DOI: 10.1029/2006JF000664.

Seroussi, H., Nakayama, Y., Larour, E., Menemenlis, D., Morlighem, M., Rignot, E., and Khazendar, A. (2017). Continued retreat of Thwaites Glacier, West Antarctica, controlled by bed topography and ocean circulation. *Geophysical Research Letters*, **44** (12), pp. 6191–6199. DOI: 10.1002/2017GL072910.

Seroussi, H. and Morlighem, M. (2018). Representation of basal melting at the grounding line in ice flow models. *The Cryosphere*, **12**, pp. 3085–3096. DOI: 10.5194/tc-12-3085-2018.

Smith-Johnsen, S., De Fleurian, B., Schlegel, N., Seroussi, H., and Nisancioglu, K. (2020). Exceptionally high heat flux needed to sustain the Northeast Greenland Ice Stream. *Cryosphere*, **14** (3), pp. 841–854. ISSN: 19940424. DOI: 10.5194/tc-14-841-2020.

Still, H., Campbell, A., and Hulbe, C. (2019). Mechanical analysis of pinning points in the Ross Ice Shelf, Antarctica. *Annals of Glaciology*, **60** (78), pp. 32–41. DOI: 10.1017/aog.2018.31.

Tsai, V. C., Stewart, A. L., and Thompson, A. F. (2015). Marine ice-sheet profiles and stability under Coulomb basal conditions. *Journal of Glaciology*, **61** (226), pp. 205–215. ISSN: 00221430. DOI: 10.3189/2015JoG14J221.

Weertman, J. (1957). On the Sliding of Glaciers. *Journal of Glaciology*, **3** (21), pp. 33–38. ISSN: 0022-1430. DOI: 10.3189/s0022143000024709.

Weertman, J. (1974). Stability of the Junction of an Ice Sheet and an Ice Shelf. *Journal of Glaciology*, **13** (67), pp. 3–11. DOI: 10.3189/S0022143000023327.

---

## Author Comment (AC2) · 25 Jan 2021

**Response to Anonymous Referee #2**

Holly Still and Christina Hulbe

January 25, 2021

We would like to thank Referee #2 for their helpful feedback on our work. The reviewer's comments are in black and our responses are in blue. Excerpts from the improved manuscript are italicised.

**Comments by Reviewer #2**

**Major comments**

**Proposed feedback.** This comment concerns lines 9-10, 329-337 and 378-384.
(1) I'm not sure I fully understand the proposed feedback: higher backstress from a pinning point is suggested to increase the ice thickness of the ice stream, thereby increasing the driving stress and basal drag which is then reflected by an increased occurrence of sticky spots at the base of the ice stream, making the ice stream less responsive. But to close the loop, the last effect has to feed back on the backstress generated by the ice rumples. How does this work?

It was incorrect to call this a feedback. We will refer to it as a 'connection' between the pinning points and grounded ice instead of as a 'feedback'.

(2) I don't understand how the proposed feedback between pinning points and basal traction of ice streams can be deduced from your experiments. In your inversion you find higher basal friction coefficients in MacIS in comparison to BIS. But if those arise due to the presence of the SCIR cannot be singled out.It could also be that it is the local ice velocity together with the ice thickness field that determines the occurrence of sticky spots in the inverted basal friction coefficient. This is not to say that it might not be possible, but I do not understand how the conclusion 'In the model, the larger basal drag acting on MacIS is itself,via regional changes in driving stress, a consequence of the coupled ice shelf and ice stream response to the SCIR.' can be drawn from the experiments presented in this study.

We agree entirely with this: "*It could also be that it is the local ice velocity together with the ice thickness field that determines the occurrence of sticky spots in the inverted basal friction coefficient.*" What we also assert is that because the thickness and velocity are mechanically connected to the pinning points, they too play a role in setting this property of the bed. What we *have* ignored is the role of the basal hydrology of the ice streams, which may just be different on MacIS than on BIS. This idea arises from the experiment, but

because the model does not simulate hydrology and because the friction coefficient is held fixed, we cannot evaluate it as directly as we would like. This is why we raised the idea in the Discussion. We have edited the Abstract (lines 7-10) to be more clear (we hope)

*We find that an ice stream located directly upstream of the pinning points, MacIS, is less responsive to their removal than the obliquely oriented BIS due to zones of locally higher basal drag acting on the main trunk of MacIS. This response is due to the larger basal drag inferred for MacIS, which may itself be a consequence of the coupled ice-shelf and ice-stream response to the pinning points.*

In the Discussion and Conclusions sections, we speculate rather than stating that the conclusion can be drawn directly from our experiments:

Line 329: *The model experiment presented here suggests a connection between pinning points and grounded ice flow that involves basal traction, and thus basal properties, upstream of the grounding line.*

Line 377: *Pinning points have been implicated as features that mediate the rate of grounding line retreat, yet their role in conditioning grounded ice flow has received less attention. In the present work, the direct effect of the SCIR on the momentum balance upstream of the grounding line is quantified and an indirect effect, via a connection involving ice stream basal traction, is suggested. This result is obtained by comparison of the relative sensitivity of the adjacent MacIS and BIS to the SCIR.*

**Basal friction adjustment and ice rise morphology.** This comment concerns lines 10-13, 345-346, 386-388 and Figure 3.
(1) In the study, after the inversion procedure, basal friction coefficients of the ice rumples are adjusted in the relaxation simulations. I suppose that this is motivated by large-scale change in the ice rumple morphology when using the inverted basal friction coefficients in the relaxation runs? This would be interesting to extend on, and add the results of the relaxation simulation in Figure 3.Also it would be interesting to see how the overall results of this study would be affected by using the initially inverted basal friction fields.

This was indeed a motivation, along with the poor agreement between the initial inversion result and basal drag inferred using the force budget method. In particular, relatively high basal friction values were assigned to upstream ice rumple nodes and zero values were assigned to the downstream ice rumples (see Fig. 1 at the end of this document). This is contrary to results obtained from the empirical force budget analysis referenced in the manuscript. Because the initially inverted friction coefficient is clearly incorrect, we decided not to pursue further analysis with the incorrect representation, and instead opted to manually tune the friction coefficient to represent all the individual ice rumples in the complex.

Yes, the initially inverted friction coefficient for the SCIR (with no manual adjustment) still reproduces the present-day flow speeds over the rumples, as it must. There is no uniquely correct inverse solution, but beyond that, we are concerned that the inverse approach is unlikely, without some manual intervention, to ever correctly represent these features. At some scale this would not matter but we are able to take advantage of manual mesh refinement to achieve a finer mesh resolution over ice rumples, so it makes sense to ensure that all of the individual ice rumples in the complex (and the associated velocity gradients and resistive stresses) are represented appropriately.

We have added the result of the relaxation simulation with the original, unadjusted friction coefficient to

Fig. 3 in the manuscript (see Fig. 3 in this document for the improved figure).

(2) Overall, a wrong morphology of the ice rumples after the relaxation simulations does not necessarily imply that the inversion produced wrong basal friction values as implies by statements in lines 10-12 and lines 368-388. It could also be that inconsistencies in the basal or surface mass balance or other factors causes a thinning, thickening, grounding or ungrounding of the ice rumples during the relaxation period. Don't get me wrong here, I think that your ad-hoc approach to correct the basal friction coefficient is ok. But I think that this should be discussed further and I'd be careful to blame the wrong morphology on the inverted friction coefficients alone. This should be extended on in the discussion.

As noted above, our concern arises in two ways, poor relaxed model morphology *and* poor agreement between our earlier force budget analysis and the initial model inversion.

We do agree that limited knowledge of other boundary conditions could also important. For example, a too large basal melt rate could also lead to ungrounding during relaxation. The basal melting parameterisation used here depends on ice thickness, and therefore the melt rate is largest at the grounding line and decays to zero in the vicinity of the ice rumples. This is comparable with melt rates inferred by satellite altimetry and works well to reproduce the observed grounding line position as well as the thickness of ice arriving to the area of the rumples from upstream.

The Supplementary Material includes a section entitled 'Improvements made to the model representation of pinning points'. We have added two paragraphs and figure to the Supplement explaining why we chose to manually adjust the friction coefficient (see Fig. 1 in this document).

*The friction coefficient initially inferred for the SCIR (described in Section 2.2) is adjusted to achieve a more realistic ice rumple area and geometry, and to better represent the basal drag inferred by Still et al. (2019). The inversion results in excessively large friction coefficient values (corresponding to $\tau_b > 400$ kPa) inferred for mesh elements on the upstream side of the SCIR complex, and zero values inferred for downstream ice rumple elements (Fig. S2). Landsat 8 imagery indicates that this representation is incorrect. Downstream rumples provide basal drag and generate surface relief comparable to the upstream rumples in the complex (implying that basal drag is greater than zero).*

*To examine sensitivity to basal friction, a range of different friction coefficient values ($\alpha = 0$ to $800$ $s^{1/2}$ $m^{-1/2}$) were assigned to the ice rumple nodes before model relaxation. A friction coefficient value that reproduces a relaxed model geometry close to present-day ice velocity and thickness, and that produces a basal drag magnitude similar to the basal drag inferred in a force budget analysis (Still et al., 2019), is used in the model experiments (i.e., $\alpha = 200$ $s^{1/2}$ $m^{-1/2}$). The manual adjustment ensured that all of the individual ice rumples in the complex (and the associated velocity gradients and resistive stresses) were represented in the model geometry (Fig. S2).*

(3) Please be more clear: It is stated that 'by extension, any parameter that is affected by the initialization procedure' is represented incorrectly in lines 12-13.What parameters do you mean? Similarly, in line 345 it is stated that ' the present work demonstrates the role of pinning points in parameter selection during model initialization.' Please explain more: What parameters are selected during the initialization procedure that are affected by the pinning points? Which role do the ice rises play? How is this shown in this study?

Lines 12-13: We mean parameters in the governing equations. What we were trying to express here is that incorrect parameter values in one area lead to incorrect parameter values in other areas (by virtue of the minimisation). Perhaps this is a tautology but in any case, it was not clear so we have removed the idea from the abstract. We have also improved lines 12-13 in the abstract:

*We also find that inversion of present-day flow and thickness for basal friction and ice softness, without feature-specific adjustment, leads to the incorrect representation of ice rumple morphology and an incorrect boundary condition at the ice base.*

The statement in line 345-346 has been improved:

*The present work demonstrates how selection of the **friction coefficient parameter for pinning point nodes** during model initialisation modifies the flow of upstream grounded ice.*

Furthermore, in line 389-390 is stated that 'The incorrect representation of pinning points also has implications for the inference of model parameters upstream of the grounding line during model initialization.' My understanding was that you did run the inversion to infer model parameters of basal shear stress and ice softness upstream of the grounding line based on the Bedmap2 geometry in which pinning points should be correctly represented. Or what do you refer to here?

That's right. We have edited the sentence to:

*Although not addressed here, this incorrect representation of pinning points during initialisation also has implications for the inference of the basal shear stress and ice softness parameter upstream of the grounding line.*

**Formulations.** Being more precise with your statements would make it easier for the reader to follow your ideas. For example in line 245 ('In general, the SCIR act to reduce longitudinal tensile stresses in grounded ice upstream of their location.'): in your next sentence you already mention that this is only partly true depending on the ice softness field used. Here you could directly go to the specific result or you should discuss why the general statement you made before(best supported by literature if it is not textbook knowledge) is actually not true in your experiments. See also comments on lines 293, 309, 319.

We have improved lines 245, 293, 309 and 319, as well as working on clarifying statements in the introduction, results and discussion sections.

We have rewritten lines 245-248 to lead the reader more directly to the point we aim to make. We note that it is generally true that flow obstructions generate compression and our results do not contradict this.

*In general, flow obstructions such as pinning points act to reduce longitudinal tensile stresses upstream of their locations and the non-local nature of the momentum balance may allow this affect to extend upstream of the grounding line. In the spatially variable $\bar{B}_{inv}$ case, the SCIR act to decrease longitudinal tensile stresses near the ice stream grounding lines, while in the uniform rate factor $\bar{B}_u$ case, the SCIR have a lesser impact on tensile stresses upstream of the grounding line (Fig. 6c and f). Downstream of the SCIR, the difference between the reference and perturbed models is complicated, with a pattern that depends on both the prescription of $\bar{B}$ and on the geometry of the embayment.*

**Figure captions.** Figure captions should give all relevant information on what is shown in the figure. For

example, sometimes grounding lines are shown, but it is not indicated if this is an observed position or the position obtained in the relaxation simulation or in the respective experiment. In addition, the appropriate grounding lines should be displayed in figures that are interpreted to show changes at the grounding line or upstream (Figs 5,6,7,8,11,12). See also specific comments to the figures.

The appropriate, modelled grounding lines have been added into Figures 5, 6, 7, 8, 9 and 12. These figures are included at the end of this document (Figs. 4-10). Figure 11 in the manuscript has the modelled grounding line displayed in (b).

We have added additional, relevant information to the following figure captions:

Figure 2. *Panel (a) shows the inverse rate factor $\bar{B}_{inv}$ in the floating part of the model domain. The value of uniform $B_u$ is shown in the colourbar of panel (a). Panels (b) and (c) focus on MacIS and BIS, showing the friction coefficient $\alpha$ and corresponding basal drag $\tau_b = -\alpha^2 N \boldsymbol{u}_b$. The grounding line is the observed position from Bindschadler et al. (2011).*

Figure 3. *Surface morphology and ice velocity for different basal friction coefficient $\alpha$ values assigned to SCIR model nodes. Along-flow surface elevation profiles in panels (a), (b) and (c) demonstrate how selection of the friction coefficient before model relaxation affects ice thickness and surface elevation for three ice rumples in the SCIR complex...*

Figure 4. *(a) Modelled ice velocity when $\alpha = 200\ s^{1/2}\ m^{-1/2}$. (b-d) The difference in ice velocity between the reference model ($\alpha = 200\ s^{1/2}\ m^{-1/2}$) and alternative relaxed models with varying $\alpha$ values assigned to the SCIR nodes. The grounding line position is the simulated position associated with the differing friction coefficient values.*

Figure 5. *The gravitational driving stress $\tau_d$ acting on the RIS and tributary ice streams with and without the SCIR. In (a-c), the simulation is initialised with $\bar{B}_{inv}$. In (d-f), the simulation is initialised with $\bar{B}_u$. In (c) and (f), a positive (negative) change indicates an increase (decrease) in $\tau_d$ after removal of the SCIR. The velocity contour lines have an interval of $100\ ma^{-1}$. Grounding line positions in (a) and (d) are obtained from the $\bar{B}_{inv}$ and $\bar{B}_u$ reference models (with SCIR). Grounding line positions in (b) and (e) are obtained from the $\bar{B}_{inv}$ and $\bar{B}_u$ perturbed models (without SCIR).*

Figure 9: *The total difference in ice speed between the steady-state, $\bar{B}_{inv}$ reference model (with SCIR) and the perturbed model (without SCIR) at various model timesteps following removal of the SCIR. (a) is the instantaneous response and (b-c) demonstrate the longer timescale adjustment of the ice-shelf and ice-stream system. Positive values indicate faster flow and negative values indicate slower flow. By a timestep of 150 years, the model has reached a new steady-state. The velocity contour lines have an interval of $100\ ma^{-1}$.*

Figure 10. *Model shear strain rates near the SCIR for (a) and (b) spatially variable ice properties ($\bar{B}_{inv}$ model) versus (c) and (d) uniform ice properties ($\bar{B}_u$ model). Grounding line positions are obtained from the reference and perturbed models.*

Figure 11. *(a) The total change in ice thickness 150 years after removal of the SCIR. Red indicates thinner ice and blue indicates thicker ice without the ice rumples. (b) Retreat of a section of the simulated MacIS grounding line ($\bar{B}_{inv}$ case) following removal of the SCIR plotted on the subglacial bed elevation (Fretwell et al., 2013).*

**Further comments**

Line 9-10: see major comment.

Please refer to our response to the major comment.

Line 10-13: see major comment.

Please refer to our response to the major comment.

Line 15: 'transient' changes in ice shelf geometry in contrast to 'persistent' changes in ice streams. I'm not sure I understand this statement as the changes in ice thickness and speed that you present in Figures 9 and 11 are visible in both, the ice shelf and the ice streams.

MacAyeal and Bindschadler Ice Streams speed up in response to removal of the SCIR, with no slow-down after the maximum velocity has been reached (60-70 years after pinning point removal). In contrast, some regions of the eastern Ross Ice Shelf initially exhibit an increase in flow speeds in response to SCIR removal, then the ice speed decreases as the ice shelf adjusts to a new steady state. This is what we meant by 'persistent' and 'transient'. This could be described more clearly and we have changed it to:

*Viewed from the perspective of change detection, we find that the ice shelf undergoes an adjustment to a new steady-state that involves an initial increase in ice speeds across the eastern ice shelf, followed by decaying flow speeds, as mass flux reduces thickness gradients in some areas and increases thickness gradients in others. Changes to ice-stream flow speeds persist without further adjustment, even without sustained grounding-line retreat.*

Figure 1: Add Echelmeyer Ice Stream as you are referring to it later on.

We first mention Echelmeyer Ice Stream when describing Fig. 9 so we have added the label to the maps in this figure.

Line 21: Another interesting study analyzing this is done by Pegler in 2018 ('Marine ice sheet dynamics: the impacts of ice-shelf buttressing').

We will add the reference to Pegler (2018) to Line 21.

*An ice shelf laterally confined within an embayment experiences reduced longitudinal tensile stress (and stretching) relative to an unconfined ice shelf due to lateral shearing where the ice flows past coastal features and islands (Sanderson, 1979; Haseloff and Sergienko, 2018; Pegler, 2018).*

Lines 24 and 61: A bit of care with the wording should be taken here. The term 'flow-buttressing' has been used previously in Furst et al. It calculates the buttressing parameter by selecting the ice flow direction as a normal direction. However, the ice flow direction can be very different to the normal direction at the grounding line which is used in Gudmundsson 2013 to calculate a buttressing parameter.

This is an excellent point. We conducted an analysis in the style of Furst et al. but did not include it here (read Still's thesis!). Line 24 has been changed to be correctly inclusive:

*Altogether, the rate of mass flux is moderated in an effect commonly referred to as the 'flow buttressing' exerted on upstream grounded ice by the ice shelf (Dupont and Alley, 2005, 2006; Gudmundsson, 2013; Fürst*

*et al., 2016).*

Lines 65 and 118: 'models'→'model configurations' as ISSM is only one model?

Changed as suggested.

Section 2.2: How is the basal friction parameter set in regions that are not grounded during the inversion but that ground during the transient forward simulations? How is basal friction treated in elements along the grounding lines?

Basal friction at the grounding line (for partially grounded model elements) is treated using the sub-element parameterisation ('SEP2') of Seroussi et al. (2014). Here, the friction coefficient for a partially grounded element is scaled to the area of grounded ice within an element (e.g., Gladstone et al., 2010; Seroussi et al., 2014) (the friction coefficient is defined at each mesh node). The friction coefficient for ice that grounds during transient simulations is specified according to the original inferred basal friction map (i.e., zero for floating ice). We do not adjust the friction coefficient downstream of the grounding line to a non-zero value (in case of grounding line advance) because the friction coefficient immediately upstream of the MacAyeal and Bindschadler Ice Stream grounding lines is already equal to zero (ice streams are low basal traction environments). It's also worth noting that no floating ice runs aground in the perturbed model simulations. We have added a statement to Section 2.1 'Ice Sheet model'.

*Both grounding-line migration and the representation of basal friction for partially floating elements (as the grounding line migrates) are treated using the sub-element parameterisation scheme ('SEP2') of Seroussi et al. (2014).*

Line 163: Why 150 years?

The rates of change in model ice volume, thickness, and speed were tracked to identify when a ∼steady state had been achieved. 150 years is when the rates of change reach 0.001%.

We have clarified this in the text (line 160):

*The steady-state reference model is perturbed by excavating the bathymetry beneath the SCIR to prevent mechanical contact between the ice and seafloor, and stepped forward for 150 years with a timestep of 1 year. By 150 years, the rate of change in ice shelf volume is 0.001%, indicating that the model has reached a new steady-state.*

Figure 2: It would be helpful to add here that also $B_u$ is shown in the colorbar of panel (a).

Changed as suggested.

Figure 3: It would be helpful to have (a) also the surface and velocity profile obtained with the inverted basal friction coefficient in the panels and (b) the magnitude of the optimized coefficient (e.g., averages along the lines). In addition, in panel (d) is the grey box showing the grounded regions in Bedmap2?

We have added surface elevation and velocity profiles obtained with the original inverted basal friction to Figure 3 as dotted line (see Fig. 3 at the end of this document).

A figure demonstrating the magnitude of the optimised friction coefficient has been added to the supplementary material following suggestion (b) above (Fig. 2 in this document).

The grey boxes in (a) to (d) show grounded sections in the steady-state model geometry. Panel (e) shows the geometry of the steady-state reference model. In the figure caption we state that: *Grey shaded boxes indicate model nodes where the ice shelf is grounded.*

Line 149: Is the same time stepping also applied in the perturbation experiments? If yes, how is the snapshot after 1 year shown in Figure 9(a) obtained?

Good point. We've improved the relevant sentence in Section 2.3 Experiment design:

*The steady-state reference model is perturbed by excavating the bathymetry beneath the SCIR to prevent mechanical contact between the ice and seafloor, and stepped forward for 150 years with a timestep of 1 year.*

Line 175: How is the morphology for the different friction coefficients obtained? I suppose that the 1000 years relaxation was run with different basal friction coefficients for the ice rumples?

This is correct. We have modified this sentence (Line 175):

*The friction coefficient assigned to ice rumple nodes is manually adjusted **before model relaxation** to reproduce both observed ice flow and rumple morphology (Fig. 3).*

And we've modified the figure caption:

*Fig. 3. Surface morphology and ice velocity for different basal friction coefficient $\alpha$ values assigned to SCIR model nodes. Along-flow surface elevation profiles in panels (a), (b) and (c) demonstrate how selection of the friction coefficient before model relaxation affects ice thickness and surface elevation for three ice rumples in the SCIR complex...*

Line 184: How does the optimized value compare to this value (see also comment on Figure 3)?

The optimized value varies spatially and ranges from 0 to 490 $s^{1/2}$ $m^{-1/2}$. The optimized values does not capture the spatial extent of the ice rumples and thus we only focus on the simulations with manual tuning of the friction coefficient for the ice rumples. A figure has been added to the Supplementary Material (see Fig. 1 in this document).

Line 192: Fig S2.

Corrected.

Figure 4: Is this an instantaneous velocity difference or a difference obtained after running the relaxation for 1000 years with the corresponding basal friction coefficient? What grounding line position is shown? If it is not shown here, it would be helpful to show the final grounding line positions after the relaxation runs to see how the ice rumple geometry is affected by the adjustment.

The figures show the difference between two steady states, with differing friction coefficients manually assigned to the SCIR (not the instantaneous difference). The grounding line position is the present-day position rather than the modelled position. Ice rumple grounded area remains unchanged in the friction

coefficient $\alpha$ =0, 200, 400, 600 models because we excavated the bathymetry around the rumples to prevent an increase in ice rumple area. Ungrounding of downstream rumples occurs in the $\alpha = 0$ model.

We have added the final grounding line positions (after relaxation) to the figure and improved the caption:

*(a) Modelled ice velocity when $\alpha$ = 200 $s^{1/2}$ $m^{-1/2}$. (b-d) The difference in ice velocity between the reference model ($\alpha$ = 200 $s^{1/2}$ $m^{-1/2}$) and alternative relaxed models with varying $\alpha$ values assigned to the SCIR nodes. The grounding line position is the simulated position associated with the differing friction coefficient values.*

Section 3.2: What is your main finding or conclusion from this comparison in relation with the later chapters? I think that it would help for the following chapters to analyze the difference and similarities between the results in light of the robustness of the results.

This section focuses on the differences between the simulations with variable and uniform $\bar{B}$.

The broad conclusions would be the same with either $\bar{B}_u$ or $\bar{B}_{inv}$, which is indeed interesting. However, there are differences over limited spatial extents that might be important for some applications, for example, how changes in pinning points might drive changes in crevasse mechanics.

$\bar{B}_{inv}$ is important for routing of ice around rumples (i.e., weaker ice in shear margins is represented, thus facilitating ice flow around an obstacle). The differences demonstrate that the inferred pattern of $\bar{B}$ near a pinning point does impact the pinning point contribution to flow resistance.

These concerns are raised in the manuscript.

Line 221-222: This statement seems to be true for the largest of the ice rumples but not for the smaller, second-largest one to the left?

Thanks for this comment. We know what is going on here but did not describe it. Pinning points lead to a smaller thickness gradient and surface slope upstream of their location, so when they are removed, the gradients and driving stress can increase upstream. Downstream, the opposite happens. This has been explained in the manuscript. We have modified Lines 221-223 to be more specific about where changes in $\tau_d$ are occurring:

*Pinning points generate locally large thickness gradients that in turn support the relatively high $\tau_d$ near the largest ice rumple (Fig. 1). This geometry is required to maintain mass flux past the obstacles. Upstream of the SCIR, relatively thicker ice with a lower surface slope leads to lower driving stresses in comparison to the configuration without the SCIR.*

Line 222-226: Since the figure does not show a grounding line position, it is hard to say, but from a rough estimate it does not look like driving stress along the grounding lines of the glaciers main trunks change significantly? It would be helpful if you (1) add the grounding lines in the Figure and (2) add the driving stress changes in Table 3. In addition, do you know what the blue spots in MacIS in the $B_u$ case are? Could they be numerical artifacts in individual mesh elements?

We have added the grounding lines to the figure (Fig. 4 in this document) and clarified the text, providing a sense of scale:

*In the perturbed model, thinning upstream of the former SCIR results in a larger thickness gradient and locally larger driving stresses immediately downstream of the grounding lines of MacIS and BIS, and in some locations, the locally larger $\tau_d$ (on the order of 10 kPa) and mass flux drive grounding line retreat (Section 3.4).*

The flux gates used for Table 3 are not in locations that would demonstrate what is described here and we are not sure how to make them useful to this cause.

The original paragraph considered only ice on the floating side of the grounding line. We have added a short paragraph describing the situation upstream of the grounding line:

*The SCIR may also affect $\tau_d$ upstream of the grounding line. Patches of relatively large decreases in the driving stress (blue patches in Fig. 5f) coincide with local lows in the bed elevation where thinner ice goes afloat in the perturbed model. Elsewhere, the differences between the two simulations are small, $<5$ kPa. The model does not simulate basal hydrology and $\alpha$ is held fixed, both of which may be variable and contribute to dynamic change in the driving stress.*

Line 227: How do you conclude that changes in flow buttressing are equal in both cases?

We were able to make this conclusion because we computed the along flow and maximum buttressing numbers of Fürst et al. (2016) using our model output ($B_{inv}$ and $B_u$, reference and perturbed model). While interesting, it did not add any additional insight into pinning point behaviour and we included only the force budget approach in the manuscript. We have removed this statement.

Line 235: Is this pattern consistent with the location of sticky spots and topographic features?

The relevant sentence: *Along the main trunk of MacIS, peaks in $-\bar{R}_{ll}$ form a 'rib-like' pattern characteristic of ice flow over sticky spots and an uneven subglacial bed topography (Fig. 6).*

Peaks in $-\bar{R}_{ll}$ are somewhat consistent with topographic lows in the Bedmap 2 subglacial topography. Peaks in the across-flow compressive stress $-\bar{R}_{tt}$ have much better correspondence with topographic lows in the Bedmap2 subglacial topography and the location of sticky spots (i.e., locally high friction coefficient values). A ridge that is visible in the surface topography (in the MODIS MOA) is also captured by $-\bar{R}_{tt}$.

The main point we aim to make is that this pattern is consistent with other studies:

Line 285 in the original manuscript: *Model initialisation results in a greater density of sticky spots on MacIS than on BIS, a result that is consistent with other inversions of observed velocities (Joughin et al., 2004; Sergienko et al., 2008), observations of ice stream surface morphology and textures (Stephenson and Bindschadler, 1990; Bindschadler and Scambos, 1991), and seismic surveys (Anandakrishnan and Alley, 1994; Luthra et al., 2016).*

Line 244: Not sure I understand this statement, Figure 8 shows particularly high lateral shear stress in this area?

We have rewritten Line 244:

*At this location, basal and lateral shear stresses make a lesser contribution to the force balance. The $+\bar{R}_{ll}$ pattern is stronger in the $B_{inv}$ case (Fig. 7).*

Line 244: 'exaggerated'→'stronger'? Since $B_{inv}$ is obtained through inversion, I would expect the velocity and ice softness field to be closer to present-day than for the ad-hoc assumption of constant $B_u$. Thus, I would think of $B_{inv}$ as the reference simulation and $B_u$ as a test case to support robustness.

Corrected as suggested.

Line 247: Fig.6→Fig.7.

Thanks, we corrected this.

Line 272: 'increase divergence downstream of their location' - this seems to depend on the ice softness and there is a large spot of decreased divergence (red) directly next to the rumples (on their western side) and downstream?

Agreed. We have improved the description of Fig. S7, beginning Line 272:

*In general, the SCIR reduce flow divergence in the region between Roosevelt Island, and the MacIS and BIS grounding line, with the pattern of $R_{tt}$ depending on the selection of $B_{inv}$ or $B_u$ (Fig. S7). Localised increases in divergence originate from individual ice rumples in the SCIR complex as ice flows over each obstacle. The SCIR also increase convergence near the outlet of Echelmeyer Ice Stream (Fig. S6), but to the south, the SCIR create a diverging geometry and transverse tensile stresses that are locally larger in comparison to the perturbed model without the ice rumples.*

Section 3.3.5: Maybe move this earlier so that you define 'sticky spots' before you discuss them in Section 3.3.2.

Good point. Section 3.3.5 (Basal drag) now follows Section 3.3.1 (The driving stress).

Figure 9: which case is shown here, $B_{inv}$ or $B_u$?

We have added this to the figure caption.

Line 285-286: It is really hard to tell from Figure 9 in which ice stream the speed increases more after 150 years. Would maybe be helpful to point to Table 3 here and add also absolute and relative speed changes along the glaciers grounding lines (and maybe move figures to the SI). In addition, it would be interesting to have an estimate of how far speed changes extend inland for both glaciers.

Excellent, thank you. We have added a reference to Table 3 and a figure to the Supplementary Material to demonstrate the greater increase in flow speed exhibited by Bindschadler Ice Stream (see Figure 2 in this document).

We have added an estimate of how far speed changes extend inland (after line 286):

*Similarly, differences in ice stream flow speeds with and without the SCIR extend further inland for BIS than for MacIS. A total speed change of $\geq 5\ ma^{-1}$ extends 280 km upstream of the BIS grounding line, and 230 km upstream of the MacIS grounding line.*

Section 3.4: I suggest to move this section before the changes in stresses are discussed to give the reader first an idea of how thickness, grounding line position and velocities change which then also makes it easier to interpret them with respect to changes in stresses.

The aim of the work was to quantify the role that the SCIR play in the momentum budget of the ice shelf and ice stream system. Thickness and velocity are consequences of this. We think that the *why* of ice sheet change is sometimes obscured or devalued relative to the *what* and the intent of organising the sections with the resistive stresses first was to insist that the *why* is important.

Line 293: Please be more precise here. How does this feedback work?

We changed the word 'feedback' to 'connection'. More details are elsewhere in this response.

Line 296-303: That you find an immediate slow-down upstream of the ice rumples is surprising and interesting to me. I think that your explanation that the initial slow will be reversed once the ice thins in the location of the ice rumples could be supported more: you could do an additional, simple experiment in which you do not only remove the ice rumples in the topography but also thin the ice at their former location so that the perturbed ice shelf is flatter (i.e., using the thickness distribution after 5 years) and then compare the instantaneous response. If the response is similar to your current 5 year response, then the initial response can most likely be linked to the initial thickness distribution in your perturbation experiment.

We agree that this is interesting. Our "explanation" is what we observe in the model simulation, as described in the manuscript. If we understand the comment correctly, the relaxation accomplishes what the reviewer is after without manual manipulation of the ice thickness.

Line 309-311: Be more specific here, what do you mean with 'the fundamental mechanisms are generic'?

We have improved lines 309-311:

*The magnitude and spatial pattern of the transient response is specific to the experimental design that intended to quantify the SCIR contribution to the mechanics of the RIS, rather than to investigate externally-forced change, such as pinning point modification due to basal melting. The fundamental mechanisms investigated here (i.e., the redistribution of stresses and the longer-timescale adjustment of ice flow and thickness), apply regardless of forcing.*

Line 310 'mechanics and dynamics'→'mechanics of Ross Ice Shelf'?

Changed as suggested.

Line 319-320: Be more precise here. A redistribution of mass from where to where? Is it large or small? And how do the pinning points affect the efficiency by pinning points?

We have added a reference to the figures that demonstrate how mass is redistributed across the RIS in response to the pinning points and changed the text.

*Altogether, the pinning points cause a redistribution of mass (Figs. 10a and 11a) and only a small increase in the total ice volume stored in the eastern RIS.*

Discussion: Discussion should be extended to include also a discussion of the model choices done here (e.g., sliding law), potential drawbacks and limitations of the methodology (e.g., assuming that present-day Ross Ice Shelf and the Siple Coast Ice Streams are in steady-state).

We have added the following two paragraphs to the Discussion section:

*The choice of friction law and its coefficients determine ice stream flow speeds and mass flux across the grounding line. The Budd-type friction law (Budd et al., 1979) used here is a common choice in ISSM applications, where it has been found to improve model stability and representation of grounding line migration (Seroussi and Morlighem, 2018). The grounding line is more sensitive to change than would be the case for other possible friction laws (Tsai et al., 2015; Brondex et al., 2017; Joughin et al., 2019).*

*The inferred basal friction coefficient $\alpha$ encapsulates the mechanical and thermal properties of the ice/bed interface in a single, spatially varying, parameter. Individual physical processes that control basal sliding (e.g., till deformabililty, presence of subglacial meltwater, bedrock bumps) are therefore hidden within the friction coefficient distribution. Without additional parameterisations to account for change in these processes over time, $\alpha$ is held fixed. Given the apparent dynamical connections between pinning points and basal traction on grounded ice (identified here and by Nias et al. (2016)), improved representation of sliding and of the processes responsible for basal friction are likely to lead to new insights into the behaviour of the coupled system.*

We have revised the description of the experiment design, two regimes with and without the SCIR, to be more clear about the motivation and about the "assumption" of steady state. We need a steady state as a reference against which to most easily compare the perturbed model. Where the steady state assumption causes a problem is the inference of parameter values during initialisation – the inversion assumes a steady state *and* we go on to hold these values fixed. We are not alone in facing this dilemma and we discussed implications but these are distributed through the Discussion, where they are relevant to specific issues, rather than in a stand-alone section.

For example, in the Discussion, we state:

Line 360: *From an ice dynamics point of view, this result is obtained at least in part because softer ice at the margin limits the transfer of resistive stresses generated by coastal features to the wider ice shelf. Because $\bar{B}$ is a fixed property of model elements set during initialisation, this particular pinning-point effect persists after the pinning point is lost, and in turn, facilitates retreat.*

Line 364: *There may be unintended consequences of fixed, constant $\bar{B}$ as relatively low total strain can change crystallographic preferred orientation, and thus $\bar{B}$, within an ice shelf or ice stream (Duval and Castelnau, 1995; Lutz et al., 2020; Jordan et al., 2020).*

Line 329-337: I'm not sure I understand this feedback, see also the main comment.

We changed the word 'feedback' to 'connection'. More details are elsewhere in this response.

Line 335-338: This sentence could be misunderstood to indicate that the studies of (van der Wel et al., 2013; Hoffman and Price, 2014) investigate a physical coupling between pinning points and ice stream basal properties (none of the studies includes dynamic ice shelves).

Indeed. These are interesting ideas that arose when *we* were re-reading those papers, after having conducted the experiments reported here. The simplest thing is to remove the references in line 336.

*The lower basal traction explains the greater responsiveness of BIS to the SCIR in the model simulations. In the real system, such a coupling could involve stress-driven changes in basal water flow and till properties*

*that in turn affect basal friction.*

Line 338: Looking into Table 3, the relative mass flux increases following SCIR removal of MacIS and BIS seem quite close when comparing it to other ice streams listed. I agree that it is interesting that BIS shows a similar and slightly higher response than MacIS which is located more directly upstream of the SCIR, but calling it a 'contrast' is maybe a bit too much.

We will replace 'contrast' with 'difference'. *The difference between MacIS and BIS sensitivity to the SCIR highlights...*

Lines 345-346: See major comment.

We should have stated that we were referring specifically to the friction coefficient inferred during model initialisation. Line 345-346 has been changed to:

*The present work demonstrates how selection of the friction coefficient parameter for pinning point nodes during model initialisation modifies the behaviour of the wider RIS.*

Conclusions: in this section it would be great if you could put your findings into a broader context, e.g., discussing the vulnerability of the SCIR in a changing climate and the implications of your work in this context.

We have added a paragraph to the discussion that considers the importance of the SCIR to long-term ice-shelf stability.

*The magnitude of the flow resistance currently provided by the SCIR is of the same order of magnitude as the flow resistance provided by the larger and more well-grounded Roosevelt, Crary and Steershead Ice Rises (Still et al., 2019). This finding alone does not reveal how important the SCIR are to RIS stability. If an unstable ice shelf configuration is required for irreversible grounding line retreat (Weertman, 1974; Schoof, 2007), then the simulations imply that the SCIR are unimportant to stability (SCIR removal results in a transition to a new steady-state), despite the relatively large flow resistance they provide. This is due to a regional redistribution of ice thickness and resistive stresses. The redistribution, which itself depends on embayment geometry, moderates the sensitivity of the coupled ice sheet–ice shelf system to the ice rumples. Similar redistributions should be expected for changes to other, individual pinning points. If stability is associated with crevasse and rift formation (Bassis and Ma, 2015; Borstad et al., 2017; Lai et al., 2020), the SCIR generate shear and tensile stresses that form crevasses and therefore the removal of pinning points may be expected to improve stability, although changes to shear stresses elsewhere may promote crevasse and rift formation in those locations. If stability is associated with ice shelf thickness (Gudmundsson et al., 2019), the SCIR cause a regional redistribution of mass with a net effect of about a 1% change in ice shelf mass, implying very little impact on long-term stability.*

Line 387-388: This could be misunderstood to mean that you did apply the feature-specific tuning during the inversion and not after the inversion. The second part of that sentence could be misinterpreted to state that the ice rumple morphology influences the overall results of this study, but this is not shown, as the results from Figs 5 to 12 are all done using the same basal friction coefficient for the ice rumples.

We agree that this is unclear and we have expanded upon this sentence:

*Naive inversion for the friction coefficient with no further evaluation or adjustment, where appropriate data are available, may lead to the incorrect representation of pinning point morphology, ice velocity, and the upstream flow resistance provided by pinning points. Over-simplifying or exaggerating the relative importance of pinning points in a model may lead to over- and under-estimates of the role of pinning points in the ice-shelf and grounding-line response to climate forcing.*

Line 388-390: see major comment.

Please refer to our response to the major comment.

Figure 10: Please also add the formerly grounded region in background of panels a and b.

The formerly grounded region is indicated in Figure 10c. The 'distance along flowline' labels along the $x$-axis are consistent between subfigures.

Fig S5: What is shown in the background of the figure?

The background is the ice thickness and we have added the missing colorbar. The content of this figure is now included in Fig. 1 in the manuscript (see Fig. 11 in this document).

**Additional figures for the Supplement**

[Figure]

Figure 1: (a) The initial friction coefficient resulting from the inversion with no manual adjustment and (b) is the friction coefficient adjusted to represent the basal drag computed by Still et al. (2019).

[Figure]

Figure 2: Comparison between the responses of MacIS and BIS to removal of the SCIR. (a) Ice stream flow speeds and (b) the change in flow speed computed at each timestep are computed as spatial averages across the main ice stream trunks. 
[revised manuscript text omitted]

---

## Referee Report (RR1)

**2nd review of 'Mechanics and dynamics of pinning points on the Shirase Coast, West Antarctica' by Holly Still and Christina Hulbe**

I would like to thank the authors for considering all the comments thoroughly! The updated manuscript addresses most of my comments from the first review. I think that the manuscript is in a good shape now except for some remaining remarks listed below. I'm supporting the publication of this manuscript after they were addressed. Line numbers refer to the revised manuscript (not the track changes version).

**Connection between pinning points and grounded bed properties**

Extending on the comment 'proposed feedback' (2). Thank you for reformulating this and making clear that this is a hypothesis.

- line 360 - 367, 423: To make this clearer for the reader, I think you should add more discussion including your points from the reply, e.g., 'The existence and strength of this effect can not not be deduced from the experiments presented here. Alternative explanantions for the different basal properties of both ice streams exist, such as differences in basal hydrology.'

**Basal friction adjustment**

Many thanks for addressing the points here! Some follow up questions:

- Figure 3: Which color corresponds to the selected value of $\alpha = 200$?

- line 196, lines 438-442: Looking at the ice rumple morphology in Figure 3 in comparison with Bedmap2 surface elevation and observed ice velocities, I find it actually hard to say that any of the lines for fixed $\alpha$ is better than the initially inferred friction coefficient for all three examples shown, except for the regions where you find zero friction in the inversions. Or did you base your statement on a different argument?

- line 197: using the force budget method to argue that the inverted basal friction is problematic could be strengthened by a discussion of the uncertainties related to inferring basal velocities in the light of findings by Bahr et al. (1994) and others.

- line 180-187: I think this comment was lost from my first review: Please add that it could also be that the inferred friction is okay but inconsistencies in the basal or surface mass balance or other factors can causes the (undesired) changes during the relaxation period.

**Further comments**

- line 2: You do not name any study in your introduction that analyses the role of pinning points using buttressing numbers and I am not aware of such a study.

- line 13: 'without feature-specific adjustemnt' $\rightarrow$ 'without feature-specific, a posteriori adjustemnt' to make clear that you did not change the inversion process.

- line 19: I'm missing some kind of implication, conclusion or outlook at the end of the abstract.

- line 27: Thanks for adressing this. To be clear here, I meant to replace 'flow-buttressing' simply with 'buttressing' as flow buttressing is only calculated in the study by Fuerst et al.

- line 32: where do the perturbations come from? Or do you mean 'a' or 'any' perturbation?

- line 43: simply 'buttressing'

- line 55: Your approach, to model a system with and without the ice rises, is very similar to the approaches by Goldberg et al., 2009 or Favier et al., 2012. Your approach is different in that it analyses in detail the stress patterns involved.

- line 71: You could add a quick overview over the structure of your manuscript here.

- 127: I'd suppose that you are using a combined velocity data set of both and not are running inversions for the two datasets (or even different points in time) individually?

- line 181: 'do not yield a realistic ice rumple geometry' → 'do not yield a realistic ice rumple geometry after the relaxation simulations.' (because the inversion itself does not affect ice geometry).

- line 224: The name of the section does not fit with the content (basal drag, driving stress included) anymore.

- line 236: That such a geometry is required to maintain the flow is stated at multiple locations. Maybe not required everywhere?

- line 240: that the change in driving stress and mass flux drives grounding line retreat is a bit unprecise here, because the stress changes you plot are with respect to steady states, so the grounding line retreat and thickness changes will have influenced the stresses as well. You could say 'are in line with'.

- line 349: What do you mean with Ross Ice Shelf stability? And what do you mean with an 'unstable ice shelf configuration is required for irreversibe grounding line retreat'? The studies by Weertman and Schoof are both based on passive ice shelves which would be equivalent to the absence of an ice shelf. Do you mean that the ice shelf is required to provide only little buttressing?

- line 357: Please explain better: I'm not sure I understand how stability is related to ice shelf thickness in the study by Gudmundsson et al. (2019)?

**References**

Bahr, D. B., Pfeffer, W. T., and Meier, M. F. (1994). Theoretical limitations to englacial velocity calculations. *Journal of Glaciology*, 40(136):509–518.

Gudmundsson, G. H., Paolo, F. S., Adusumilli, S., and Fricker, H. A. (2019). Instantaneous antarctic ice sheet mass loss driven by thinning ice shelves. *Geophysical Research Letters*, 46(23):13903–13909.

---

## Author Response (AR2)

**Response to Anonymous Referees #1 and #2**

**Holly Still and Christina Hulbe**

**April 1, 2021**

We would like to thank both referees for providing additional feedback to improve the revised manuscript. The reviewers' comments are in black and our responses are in blue. Excerpts from the improved manuscript are italicised.

**Comments from Reviewer #1**

Comment related to original line 118-125 "grid resolution". I would still discourage the use of "resolution". If you don't like "posting" how about "grid spacing".

We have changed resolution to grid-spacing when referring to interpolated data sets.

Line 149 comment. Perhaps I am missing something, and it's a minor nit, but if you have edge lengths of 500 m, then you can resolution down to ∼500m. For example, you could have 4 points forming a 500x500m square (2-triangles) in your mesh, which would be somewhat equivalent to a 500-m regular grid. So in that sense it "ranges from 500m to ..." Not 1000 m.

We have clarified that this is an "unstructured triangular mesh" and improved the relevant sentence:

*"Mesh element size ranges from approximately 1 km edge lengths over pinning points, ice-stream shear margins and the grounding line, to approximately 10 km edge lengths over the inland ice sheet and central ice shelf."*

**Comments from Reviewer #2**

**Connection between pinning points and grounded bed properties**

Extending on the comment 'proposed feedback' (2). Thank you for reformulating this and making clear that this is a hypothesis.

Line 360 - 367, 423: To make this clearer for the reader, I think you should add more discussion including your points from the reply, e.g., 'The existence and strength of this effect can not not be deduced from the experiments presented here. Alternative explanations for the different basal properties of both ice streams exist, such as differences in basal hydrology.'

We think that our writing is clear (thank you for the positive comment about the writing) and agree that a bit more could be said. While different basal materials or states would not discount the hypothesis, they would make it more complicated to investigate. We have added the following sentence:

'*A model in which basal till properties are coupled to the basal hydrologic system would be required to investigate this hypothesis further.*"

**Basal friction adjustment**

Figure 3: Which color corresponds to the selected value of $\alpha = 200$?

Excellent point. The $\alpha = 200$ lines have been identified using a label on the figure.

Line 196, lines 438-442: Looking at the ice rumple morphology in Figure 3 in comparison with Bedmap2 surface elevation and observed ice velocities, I find it actually hard to say that any of the lines for fixed $\alpha$ is better than the initially inferred friction coefficient for all three examples shown, except for the regions where you find zero friction in the inversions. Or did you base your statement on a different argument?

We agree that it would not be sensible to use only these profiles to make a decision. As reported in the text, we used a regional, 2-D, evaluation and a comparison with results from a force budget method. It is not practical to show all of those maps in a figure and we think that the suites of profiles in Figure 3 show what is happening, mechanically, among the coupled variables as adjustments are made to $\alpha$. This fits with the point of view and objectives of the study and the paper.

Line 197: using the force budget method to argue that the inverted basal friction is problematic could be strengthened by a discussion of the uncertainties related to inferring basal velocities in the light of findings by Bahr et al. (1994) and others.

We rely everywhere on an inversion using the shallow-shelf equations. Basal parameters found in this way are similar to those found by inversion of full-Stokes equations, except where ice thickness gradients are relatively large and the SSA assumption is too simple (Morlighem et al. (2010), comparison of SSA and full-Stokes inversions).

The ice rumples are such a location. Comparing basal drag resulting from the two approaches, the initial ISSM inversion gives an average basal drag of 14 kPa for the SCIR elements (most of which have $\tau_b = 0$) while the force budget approach gives 51.6$\pm$18.3 kPa. The scale of the difference is similar to differences across approaches in Morlighem et al. (2010). A reasonable conclusion is that the FS equations should be used not only at grounding lines, as those authors suggest, but also over ice rumples.

We have added to the Discussion, line 401: *"Different momentum equations, for example, full-Stokes instead of the SSA, will also yield different results (Morlighem et al., 2010), particularly where thickness gradients are large, and this may explain the difference between our ISSM-inversion and force budget inferences of the basal friction parameter."*

Line 180-187: I think this comment was lost from my first review: Please add that it could also be that the inferred friction is okay but inconsistencies in the basal or surface mass balance or other factors can causes the (undesired) changes during the relaxation period.

The inferred friction was undesirable before it had undesirable effects on the ice thickness during relaxation. We do agree that limited knowledge of other boundary conditions does not help this situation. For example, a too large basal melt rate could also lead to ungrounding during relaxation.

**Further comments**

Line 2: You do not name any study in your introduction that analyses the role of pinning points using buttressing numbers and I am not aware of such a study.

In the Introduction, line 43 we stated: *Flow buttressing numbers (Gudmundsson, 2013; Fürst et al., 2016) provide a summary view of the non-local effects but do not quantify the pinning point contribution to individual resistive stresses.*

In their Supplementary Material, Fürst et al. (2016) compute along-flow and $\sigma_2$ buttressing numbers for the Larsen C Ice Shelf. They simulate the 'ungrounding' of Bawden Ice Rise by setting the basal friction coefficient to zero at that location, and they find that loss of the ice rise increases the area of passive shelf ice (i.e., modifies the maximum $\sigma_2$ buttressing number near the shelf front).

Additional authors who compute buttressing numbers with pinning points included in their model domains and acknowledge that pinning points contribute to the flow buttressing pattern:

- Kingslake et al. (2018) compute flow buttressing numbers using the method of Fürst et al. (2016) to assess the relevance of buttressing associated with ice rise formation in the Ross and Weddell Sea sectors.

- Borstad et al. (2013) consider the consequences of the Larsen C Ice Shelf losing contact with Bawden Ice Rise. They compute a buttressing parameter $f$ (a normalised measure of the backstress introduced by Dupont and Alley (2005)) and find that Bawden and Gipps Ice Rises contribute locally high backstress which they describe as 'overbuttressed' with $f > 1$ (compressive flow).

While Gudmundsson (2013) provide a good overview of buttressing number theory, pinning points are not the focus of this paper. We will swap this for a reference that specifically computes buttressing numbers and considers the pinning point contribution. *Flow buttressing numbers (Borstad et al., 2013; Fürst et al., 2016) provide a summary view of the non-local effects but do not quantify the pinning point contribution to individual resistive stresses.*

Line 13: 'without feature-specific adjustment' → 'without feature-specific, a posteriori adjustment' to make clear that you did not change the inversion process.

Updated, thank you.

Line 19: I'm missing some kind of implication, conclusion or outlook at the end of the abstract.

We have moved a sentence to the end of the Abstract and edited it slightly. The sentence *"Where pinning point effects are important, model tuning that respects pinning point morphology is necessary to represent the ice-sheet and ice-shelf system as a whole"* has been changed to:

*Where pinning point effects are important, model tuning that respects their morphology is necessary to represent the system as a whole and inform interpretations of observed change.*

Line 27: Thanks for addressing this. To be clear here, I meant to replace 'flow-buttressing' simply with 'buttressing' as flow buttressing is only calculated in the study by Furst et al.

We have changed 'flow-buttressing' to 'buttressing'.

*Altogether, the rate of mass flux is moderated in an effect commonly referred to as 'buttressing' exerted on upstream grounded ice by an ice shelf (Dupont and Alley, 2005, 2006; Gudmundsson, 2013; Fürst et al., 2016).*

Line 32: where do the perturbations come from? Or do you mean 'a' or 'any' perturbation?

Thank you, we have changed this to:

*Any momentum and mass perturbation must be balanced by changes in thickness and resistive stresses elsewhere in the ice-shelf and ice-sheet system.*

Line 43: simply 'buttressing'

Corrected.

Line 55: Your approach, to model a system with and without the ice rises, is very similar to the approaches by Goldberg et al., 2009 or Favier et al., 2012. Your approach is different in that it analyses in detail the stress patterns involved.

Line 55: *"We take a different approach to examine the dynamical role of pinning points."*

Has been changed to:

*"We take a different perspective to examine the dynamical role of pinning points that involves a detailed analysis of the stress patterns across the RIS. Our aim is to quantify..."*

Line 71: You could add a quick overview over the structure of your manuscript here.

We have thought about this but the paper has a very standard structure and we could not find a way to do it that did not seem repetitive.

127: I'd suppose that you are using a combined velocity data set of both and not are running inversions for the two datasets (or even different points in time) individually?

We have added an additional statement:

*Surface velocities are from the 750 m grid-spacing Landsat 8 dataset (Fahnestock et al., 2016) and the 900 m grid-spacing MEaSUREs dataset (Rignot et al., 2011) representing time periods from 2013 to 2016, and 2007 to 2009, respectively. The two velocity datasets are merged, with the MEaSUREs dataset used to fill the region beyond the Landsat 8 latitudinal limit.*

Line 181: 'do not yield a realistic ice rumple geometry' → 'do not yield a realistic ice rumple geometry after the relaxation simulations.' (because the inversion itself does not affect ice geometry).

Corrected.

Line 224: The name of the section does not fit with the content (basal drag, driving stress included) anymore.

We have changed the original section subheading '*3.3 Partitioning of resistive stresses*' to '*3.3 Stresses*'.

Line 236: That such a geometry is required to maintain the flow is stated at multiple locations. Maybe not required everywhere?

We have deleted the statement '*This geometry is required to maintain mass flux past the obstacles*' in Section 3.3.1. We have left the idea in Section 3.1 where it is first introduced.

Line 240: that the change in driving stress and mass flux drives grounding line retreat is a bit unprecise here, because the stress changes you plot are with respect to steady states, so the grounding line retreat and thickness changes will have influenced the stresses as well. You could say 'are in line with'.

Thanks, we have made the suggested change.

Line 349: What do you mean with Ross Ice Shelf stability? And what do you mean with an 'unstable ice shelf configuration is required for irreversible grounding line retreat'? The studies by Weertman and Schoof are both based on passive ice shelves which would be equivalent to the absence of an ice shelf. Do you mean that the ice shelf is required to provide only little buttressing?

We have replaced 'Ross Ice Shelf stability' with 'the present-day configuration of the RIS'.

We meant for these references to apply only to MISI type retreat but agree that this is unclear. The paragraph has been re-written:

*This finding alone does not reveal how important the SCIR are to maintaining the present day configuration of the RIS. If stability is associated with grounding line position, then the simulations imply that the SCIR are unimportant (SCIR removal results in a transition to a new steady-state), despite the relatively large flow resistance they provide. This is due to a regional redistribution of ice thickness and resistive stresses. The redistribution, which itself depends on embayment geometry, moderates the sensitivity of the coupled ice-sheet and ice-shelf system to the ice rumples. Similar redistributions should be expected for changes to other, individual pinning points. If stability is associated with crevasse and rift formation (Bassis and Ma, 2015; Borstad et al., 2017; Lai et al., 2020), the SCIR generate shear and tensile stresses that form crevasses and therefore the removal of pinning points may be expected to improve stability, although changes to shear stresses elsewhere may promote crevasse and rift formation in those locations.*

Line 357: Please explain better: I'm not sure I understand how stability is related to ice shelf thickness in the study by Gudmundsson et al. (2019)?

We thought that Gudmundsson et al. (2019) was quite relevant:

"Our results have important implications for assessing future mass loss from the Antarctic ice sheet and resulting sea-level rise: Thinning of ice shelves is now causing a significant increase in discharge of grounded ice into the oceans; because this process is almost instantaneous, we are not protected against the impact of the Antarctic ice sheet on global sea levels by a long response time." (Gudmundsson et al., 2019)

We revised and simplified the paragraph on stability and the Gudmundsson et al. (2019) sentence and reference have been removed.

**Modified figures**

[Figure]

Figure 1: Surface morphology and ice velocity for different basal friction coefficient $\alpha$ values assigned to SCIR model nodes. Along-flow surface elevation profiles in panels (a), (b) and (c) demonstrate how selection of the friction coefficient before model relaxation affects ice thickness and surface elevation for three ice rumples in the SCIR complex (ice rumples A, B and C, respectively, see Fig. 1b for their location). Grey shaded boxes indicate model nodes where the ice shelf is grounded. Panel (d) demonstrates how selection of the friction coefficient affects the velocity magnitude. The profile in (d) represents a single pathway that begins 150 km upstream of the MacIS grounding line, intersects the SCIR rumple C, and ends at the shelf front. Panel (e) shows ice thickness and the underlying seafloor along this pathway in the reference model. The locations of the profiles in (a) to (e) are mapped in Fig. 1b.

**References**

Bassis, J. N. and Ma, Y (2015). Evolution of basal crevasses links ice shelf stability to ocean forcing. *Earth and Planetary Science Letters*, **409**, pp. 203–211. DOI: 10.1016/j.epsl.2014.11.003.

Borstad, C. P., Rignot, E., Mouginot, J., and Schodlok, M. P. (2013). Creep deformation and buttressing capacity of damaged ice shelves: theory and application to Larsen C ice shelf. *The Cryosphere*, **7** (6), pp. 1931–1947. DOI: 10.5194/tc-7-1931-2013.

Borstad, C., Mcgrath, D., and Pope, A. (2017). Fracture propagation and stability of ice shelves governed by ice shelf heterogeneity. *Geophysical Research Letters*, **44**, pp. 4186–4194. DOI: 10.1002/2017GL072648.

Dupont, T. K. and Alley, R. B. (2005). Assessment of the importance of ice-shelf buttressing to ice-sheet flow. *Geophysical Research Letters*, **32** (4), p. L04503. DOI: 10.1029/2004GL022024.

Fahnestock, M., Scambos, T., Moon, T., Gardner, A., Haran, T., and Klinger, M. (2016). Rapid large-area mapping of ice flow using Landsat 8. *Journal of Geophysical Research: Earth Surface*, **121** (2), pp. 283–293. DOI: 10.1016/j.rse.2015.11.023.

Fürst, J. J., Durand, G., Gillet-Chaulet, F., Tavard, L., Rankl, M., Braun, M., and Gagliardini, O. (2016). The safety band of Antarctic ice shelves. *Nature Climate Change*, **6** (5), pp. 479–482. DOI: 10.1038/nclimate2912.

Gudmundsson, G. H. (2013). Ice-shelf buttressing and the stability of marine ice sheets. *The Cryosphere*, **7** (2), pp. 647–655. DOI: 10.5194/tc-7-647-2013.

Gudmundsson, G. H., Paolo, F. S., Adusumilli, S., and Fricker, H. A. (2019). Instantaneous Antarctic ice sheet mass loss driven by thinning ice shelves. *Geophysical Research Letters*, **46** (23), pp. 13903–13909. ISSN: 0094-8276. DOI: 10.1029/2019GL085027.

Kingslake, J., Scherer, R. P., Albrecht, T., Coenen, J., Powell, R. D., Reese, R., Stansell, N. D., Tulaczyk, S., Wearing, M. G., and Whitehouse, P. L. (2018). Extensive retreat and re-advance of the West Antarctic Ice Sheet during the Holocene. *Nature*, **558** (7710), pp. 430–434. DOI: 10.1038/s41586-018-0208-x.

Lai, C.-Y., Kingslake, J., Wearing, M., Po-Hsuan Chen Cameron, Gentine, P., Li, H., Spergel, J., and Wessem, J. M. van (2020). Vulnerability of Antarctica's ice shelves to meltwater-driven fracture. *Nature*, **584**, pp. 574–578. DOI: 10.1038/s41586-020-2627-8.

Morlighem, M., Rignot, E., Seroussi, H., Larour, E., Ben Dhia, H., and Aubry, D. (2010). Spatial patterns of basal drag inferred using control methods from a full-Stokes and simpler models for Pine Island Glacier, West Antarctica. *Geophysical Research Letters*, **37** (14), p. L14502. DOI: 10.1029/2010GL043853.

Rignot, E, Mouginot, J, and Scheuchl, B (2011). MEaSUREs InSAR-Based Antarctica Ice Velocity Map. DOI: 10.5067/measures/cryosphere/nsidc-0484.001.

---

## Author Response (AR3)

**Reply to comments**

Holly Still and Christina Hulbe

April 18, 2021

Dear Olivier,

Thank you for the additional feedback. The editor's comments are in black and our responses are in blue. Excerpts from the improved manuscript are italicised.

**Comments**

The caption of Fig. 1 is not precise enough: In (a), you should mention to what refer the white dashed line (limit of the FE model domain?). I would suggest to add a square in (a) showing the limit of (b). Also, "grounding zone" should be "grounding lines". In (b), you should specify what is the white dashed line and yellow lines.

We have added additional detail about the white dashed line in (a) and the cross-section lines in (b). 'Grounding zone' has been replaced with 'grounding lines'. We tried adding an inset square to (a) and we found that it made the map too cluttered. Instead, the SCIR label shows the overlapping spatial extent between (a) and (b).

The improved figure caption:

*Pinning points in the RIS. In panel (a), large pinning points are labelled: SCIR = the Shirase Coast Ice Rumples, RI = Roosevelt Island, SIR = Steershead Ice Rise and CIR = Crary Ice Rise. The colour map of surface ice velocity magnitude is from the MEaSURES velocity dataset (Rignot et al., 2011). The black line indicates the grounding lines (Bindschadler et al., 2011) and the white dashed line is the limit of the finite element model domain. Panel (b) shows the along-flow cross-sections intersecting the SCIR in Figs. 3 and 10 (yellow and white dashed lines), and the gates used for mass flux calculations in Table 3. The colour map of ice thickness is from the Bedmap2 compilation (Fretwell et al., 2013). In each figure from hereon, datasets are mapped with a Polar Stereographic Projection with a central meridian of 0° and a standard latitude of 71°S, and in most cases, overlayed onto the MODIS MOA (Haran et al., 2014).*

In Fig. 1, on the north part of the shelf, the model domain seems to cut the ice-streams. Why this choice? What type of BC do you apply at the model domain boundary (observed velocity?)?

To the east/north-east of the Ross Ice Shelf, the domain follows the topographic divide delineating the

drainage basin of the West Antarctic Ice Sheet ice streams (mapped by Zwally et al. (2012)). To the west of the RIS, the domain is constrained by the Transantarctic Mountains.

We have added a sentence on the boundary conditions to the manuscript (Line 134):

*Dirichlet conditions are imposed on the upstream boundaries of the model domain using observed velocity and ice thickness, and zero-slope Neumann conditions are specified on the downstream (ice shelf front) boundary.*

line 161: 1000 years is a very long relaxation? Model might reach some steady state already after such a long relaxation period? Usually relaxation periods are of few years to few decades?

Because this is a perturbation experiment, the aim was to achieve as close to zero transients as possible during initialisation. This required centuries rather than decades.

Line 321: Negative response of which variable? (The mean instantaneous velocity change over the whole ice-stream is, however, negative?)

Thanks, we have made the suggested change:

*The mean instantaneous velocity change over the lower MacIS and BIS is, however, negative (Fig. 9a).*

Caption Fig. 9: Positive values indicate faster flow for the perturbed model without SCIR and ...

Thanks, we have made the suggested change to the figure caption:

*The total difference in ice speed between the steady-state, $\bar{B}_{inv}$ reference model (with SCIR) and the perturbed model (without SCIR) at various model timesteps following removal of the SCIR. (a) is the instantaneous response and (b-c) demonstrate the longer timescale adjustment of the ice-shelf and ice-stream system. Positive values indicate faster flow for the perturbed model without the SCIR and negative values indicate slower flow. By a timestep of 150 years, the model has reached a new steady-state. Grounding-line positions are from the perturbed model at timesteps of 1, 5 and 150 years. The velocity contour lines have an interval of 100 $ma^{-1}$*

Line 447: may be also conclude that an inversion alone is not sufficient for transient prognostic simulations and that appropriate evolution equations for the ice rheology and the basal friction would be required to have a correct evolution of these fields induced by evolving flow conditions?

Thanks, we have added two additional sentences to follow Line 447:

*The present contribution demonstrates the importance of high fidelity representation of pinning points for simulation of their effects in system models and, by extension, how observed change is interpreted. The use of spatially- and temporally-fixed ice properties and basal friction may also be insufficient to represent a changing pinning point environment. Prognostic simulations of pinning points dynamics will be further improved by the implementation of process-based ice properties and basal friction fields that respond to changing flow conditions.*

**References**

Bindschadler, R, Choi, H, Wichlacz, A, Bingham, R, Bohlander, J, Brunt, K, Corr, H, Drews, R, Fricker, H, Hall, M, Hindmarsh, R, and Kohler, J (2011). Getting around Antarctica: new high-resolution mappings of the grounded and freely-floating boundaries of the Antarctic ice sheet created for the International Polar Year. *The Cryosphere*, **5** (3), pp. 569–588. DOI: 10.5194/tc-5-569-2011.

Fretwell, P. et al. (2013). Bedmap2: improved ice bed, surface and thickness datasets for Antarctica. *The Cryosphere*, **7** (1), pp. 375–393. DOI: 10.5194/tc-7-375-2013.

Haran, T., Bohlander, J., Scambos, T., Painter, T., and Fahnestock, M. (2014). MODIS Mosaic of Antarctica 2008-2009 (MOA2009) Image Map. DOI: 10.7265/N5KP8037.

Rignot, E, Mouginot, J, and Scheuchl, B (2011). MEaSUREs InSAR-Based Antarctica Ice Velocity Map. DOI: 10.5067/measures/cryosphere/nsidc-0484.001.

Zwally, H. J., Giovinetto, M. B., Beckley, M. A., and Saba, J. L. (2012). Antarctic and Greenland Drainage Systems [Dataset].